# Stable Localized Conformal Prediction via Transduction

**Yinjie Min** [1]  **Liuhua Peng** [2]  **Changliang Zou** [1]

## Abstract

Existing evaluations of conformal prediction, such as prediction efficiency and test-conditional coverage, are defined in expectation over the calibration data. In practice, when only one calibration set of limited size is available, prediction sets often exhibit high variability in size, especially for methods with localization. We formalize this concern as *set stability*, defined as the variance of the conditional expectation of the set size given the calibration data. To improve stability without requiring additional target-task labels, we propose Stable Conformal Prediction (StCP), a transfer learning approach that utilizes labeled source-task data and unlabeled target data. Theoretically, we characterize the marginal coverage and stability of StCP; empirically, it delivers more stable prediction sets than standard conformal prediction methods, especially for those with localization, when calibration data are limited.

## 1. Introduction

Machine learning algorithms typically provide point predictions, such as conditional mean estimates for regression or class labels for classification. In risk-sensitive applications, however, reliable uncertainty quantification is essential to ensure safety (Kompa et al., 2021). Conformal prediction (CP) (Vovk et al., 2005) offers a statistically rigorous approach that outputs set-valued predictions with guaranteed coverage. In the standard formulation, we are given a labeled calibration dataset $\mathcal{L}_n = \{(X_i, Y_i)\}_{i=1}^n$, where the labeled pairs are drawn i.i.d. from a distribution $P = P_X \times P_{Y|X}$ supported on $\mathcal{X} \times \mathcal{Y}$. Using $\mathcal{L}_n$, the goal is to construct a prediction set $\widehat{C}(\cdot)$ such that for a new test point $(X_{n+1}, Y_{n+1}) \sim P$ with $Y_{n+1}$ unobserved, the marginal coverage probability satisfies

$$\mathbb{P}\big(Y_{n+1} \in \widehat{C}(X_{n+1})\big) \geq 1 - \alpha,$$

where $1 - \alpha \in (0, 1)$ is a user-specified nominal coverage level. This finite-sample, distribution-free coverage guarantee constitutes a foundation of conformal prediction.

Beyond marginal coverage, existing conformal prediction methods are often evaluated from two perspectives (Zhou et al., 2025). First, the *prediction efficiency* (Vovk et al., 2016), often measured by the expected size of the prediction set $\mathbb{E}|\widehat{C}(X_{n+1})|$, where $|\cdot|$ denotes the Lebesgue measure, quantifies its precision. Among all sets satisfying the marginal coverage requirement, shorter or smaller sets are more efficient and informative, and thus preferred. Second, *test-conditional coverage* (Shafer & Vovk, 2008) is considered because a marginally valid prediction set may still undercover in certain regions of the data space. The test-conditional coverage at a point $x \in \mathcal{X}$ is defined as $\mathbb{P}(Y_{n+1} \in \widehat{C}(X_{n+1}) \mid X_{n+1} = x)$. It has been shown that strict finite-sample conditional coverage is often unattainable without strong assumptions (Lei et al., 2013; Lei & Wasserman, 2014). Therefore, existing works aim for it to be asymptotically close to the required level $1 - \alpha$.

While both criteria provide important perspectives on conformal prediction sets, they are defined in expectation over the calibration data $\mathcal{L}_n$ and therefore primarily characterize average behavior across repeated calibration samples. In practice, however, we typically observe only one calibration dataset. Consequently, a method with good average performance may still produce prediction sets with highly variable sizes for a given calibration sample, which is a key practical concern not captured by expectation-based metrics. This calibration-conditional perspective is related to works on training-conditional coverage (Bian & Barber, 2023; Liang & Barber, 2025), which study how coverage depends on the calibration sample. In this paper, we focus on the stability of prediction set size under calibration randomness.

The issue of highly variable set sizes is particularly pronounced for conformal prediction methods that target asymptotic test-conditional coverage (Lei & Wasserman, 2014; Romano et al., 2019; Chernozhukov et al., 2021; Guan, 2023; Gibbs et al., 2025). These methods typically rely on local calibration around the test point, and we generically refer to them as *conformal prediction with localiza-*

[1]School of Statistics and Data Science, Nankai University, Tianjin, China [2]School of Mathematics & Statistics, The University of Melbourne, Melbourne, Australia. Correspondence to: Liuhua Peng <liuhua.peng@unimelb.edu.au>, Changliang Zou <nk.chlzou@gmail.com>.

*Proceedings of the 43rd International Conference on Machine Learning*, Seoul, South Korea. PMLR 306, 2026. Copyright 2026 by the author(s).

*tion* methods. Even with a well-trained predictive model, limited calibration sample size can induce substantial instability in prediction sets. The local calibration step amplifies fluctuations from small effective sample sizes, making the output highly sensitive to the realized calibration set, which we further analyze in Section 2.

These observations motivate the need for a novel evaluation metric that quantifies the sensitivity to calibration data. We introduce **set stability**, defined as the variance of the expected prediction set size with respect to the calibration data. Since the conformal prediction set $\widehat{C}(\cdot)$ is constructed from the calibration data, its properties align closely with this dataset. Formally, let $\mathcal{D}$ represent all data required to construct prediction set $\widehat{C}(\cdot)$, including the calibration data $\mathcal{L}_n$. The conditional expected size of the prediction set given $\mathcal{D}$ is defined as $L(\widehat{C}) = \mathbb{E}\{|\widehat{C}(X_{n+1})| \mid \mathcal{D}\}$, where the expectation is taken over the test covariate $X_{n+1}$. Crucially, this expectation over $X_{n+1}$ averages out test-point randomness, so $L(\widehat{C})$ reflects the average set size given calibration dataset. The set stability is then measured by

$$\mathrm{Var}(L(\widehat{C})) = \mathrm{Var}(\mathbb{E}\{|\widehat{C}(X_{n+1})| \mid \mathcal{D}\}). \qquad (1)$$

When $\mathcal{D} = \mathcal{L}_n$, this quantity captures the variability induced by the randomness in calibration data. A small variance indicates that the prediction set is robust to a specific calibration sample, thus providing more stable performance in practical one-shot scenarios.

To illustrate set stability, Figure 1 shows that both prediction efficiency and stability improve as the calibration size increases. In practice, however, the available calibration data are often limited due to time and cost constraints. Fortunately, unlabeled data from $P_X$ are typically more accessible in many real-world scenarios (Van Engelen & Hoos, 2020). Additionally, labeled data from related tasks may also be available, such as data from different but related tasks as in transfer learning and meta-learning (Pan & Yang, 2009; Hospedales et al., 2021), or data from other agents as in federated learning (Mammen, 2021). One motivating example is tissue classification for medical risk stratification used in Section 5: low-risk patients often contribute abundant, well-labeled data, while high-risk cases are rarer and may have delayed or uncertain labels due to limited follow-up or diagnostic ambiguity, resulting in scarce labeled target data but relatively abundant unlabeled target covariates.

Therefore, we consider a weakly-supervised setting that incorporates auxiliary information from both target and source tasks. Formally, in addition to the labeled calibration set $\mathcal{L}_n$ drawn from $P$, we assume access to a larger unlabeled target sample $\mathcal{U}_m = \{\widetilde{X}_j\}_{j=1}^m$ drawn from $P_X$, as well as abundant labeled source data $\mathcal{L}'_N = \{(X'_k, Y'_k)\}_{k=1}^N$ drawn i.i.d. from $P' = P'_X \times P'_{Y|X}$, with $N \wedge m \gg n$. We assume transfer learning is feasible: while the covariate distribu-

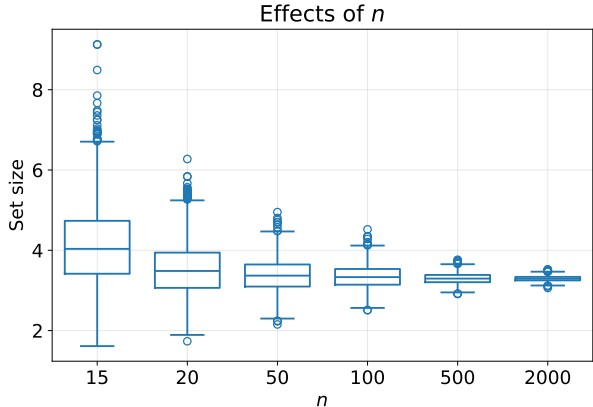

*Figure 1.* Effects of $n$ on the prediction efficiency and stability.

tions $P_X$ and $P'_X$ may differ substantially, the conditional distributions $P_{Y|X}$ and $P'_{Y|X}$ are similar.

Under this setup, we propose a general transductive framework to stabilize conformal prediction while preserving core predictive properties. Our contributions are as follows:

(i) We propose Stable Conformal Prediction (StCP), a general stabilization framework for conformal prediction that transfers source-task information and leverages unlabeled target data. Since instability is more pronounced under localization, StCP brings greater improvement in localized settings, yielding stable localized conformal prediction (SLCP) with enhanced stability.

(ii) We establish the marginal coverage guarantee of StCP and theoretically demonstrate the improvement of its set stability via transduction.

(iii) We validate StCP on real-world datasets, demonstrating that it improves set stability while maintaining satisfactory marginal and test-conditional performance compared with baseline methods.

### 1.1. Related Works

**Test-conditional coverage:** Since exact test-conditional coverage is theoretically unattainable under weak assumptions (Lei et al., 2013), researchers have developed conformal methods that target asymptotic conditional coverage. These typically operate via localized calibration (Lei & Wasserman, 2014; Guan, 2023; Hore & Barber, 2025; Gibbs et al., 2025) or via more informative score functions (Romano et al., 2019; Chernozhukov et al., 2021) to adapt prediction sets locally. A recent line of work (Min et al., 2026) further unifies and extends these approaches.

**Prediction efficiency:** To improve prediction efficiency of conformal prediction sets, Yang & Kuchibhotla (2025) and Liang et al. (2024) integrate parameter selection into cal-

ibration to obtain smaller sets while preserving marginal coverage validity, whereas Kiyani et al. (2024) directly optimizes set length via constrained minimization.

**Stability:** Stability has been extensively studied in learning theory as a measure of sensitivity to data perturbations and a tool for understanding generalization (Bousquet & Elisseeff, 2002; Hardt et al., 2016), and has also been used in unsupervised learning to assess robustness and guide model selection (Von Luxburg, 2010).

**Semi-supervised conformal prediction:** Recent work has explored leveraging unlabeled data to improve conformal prediction when calibration size is small. Seedat et al. (2023) employs unlabeled data to train a more adaptive conformal normalization model. Based on prediction-powered inference (PPI) (Angelopoulos et al., 2023), Einbinder et al. (2025) tunes risk-controlling parameters using imputed labels from unlabeled data. Angelman et al. (2025) addresses source-free domain adaptation settings by using pseudo-labels for calibration. Meanwhile, Wen et al. (2025) develops a semi-supervised distribution learning framework that estimates the entire cumulative distribution function with both labeled and unlabeled data.

**Leveraging auxiliary information:** For scenarios with limited calibration data, many methods leverage auxiliary information. Fisch et al. (2021) uses meta-learning to transfer information from auxiliary tasks, achieving asymptotic marginal guarantees on the target. Bashari et al. (2025) employs simulated data to enhance calibration, yet its guarantees rely on distributional similarity. In personalized federated settings, Min et al. (2026) utilizes other agents to improve test-conditional coverage for a target agent.

## 2. Formulation and Motivation

We start from a generic conformal construction with a pre-trained non-conformity score $S : \mathcal{X} \times \mathcal{Y} \to \mathbb{R}$. The score may be an ordinary residual score or a score from localized methods, such as the GLCP score discussed in Remark 2.1.

Given the score, the conformal set is constructed as

$$\widehat{C}(X_{n+1}) = \{y : S(X_{n+1}, y) \le \widehat{q}\}, \qquad (2)$$

with $\widehat{q} = Q(1 - \alpha; (n+1)^{-1}\{\sum_{i=1}^{n} \delta_{S(X_i, Y_i)} + \delta_\infty\})$ defined as the $1 - \alpha$ quantile of the empirical distribution $(n+1)^{-1}\{\sum_{i=1}^{n} \delta_{S(X_i, Y_i)} + \delta_\infty\}$, where $\delta_a$ is a point mass at $a$. As $(X_1, Y_1), \ldots, (X_n, Y_n), (X_{n+1}, Y_{n+1})$ are i.i.d., the set achieves the marginal coverage guarantee

$$\mathbb{P}(Y_{n+1} \in \widehat{C}(X_{n+1})) \in \left[1 - \alpha, 1 - \alpha + (n+1)^{-1}\right].$$

When the pre-trained score $S$ is sufficiently informative, the set $\widehat{C}(X_{n+1})$ can achieve asymptotic test-conditional coverage (e.g., in localized methods (Min et al., 2026)):

$$\mathbb{P}(Y_{n+1} \in \widehat{C}(X_{n+1}) \mid X_{n+1} = x) \to 1 - \alpha$$

for any fixed $x \in \mathcal{X}$ as $n \to \infty$.

*Remark* 2.1. As a special case, GLCP (Min et al., 2026) sets $S(x, y) = \phi(\widehat{F}_{V|X}(V(x, y) \mid x))$ with a base score $V(x, y)$ and a fixed map $\phi : [0, 1] \to [0, 1]$. When $V(x, y) = |y - \widehat{\mu}(x)|$ and $\phi(v) = v$, where $\widehat{\mu}(\cdot) : \mathcal{X} \to \mathcal{Y}$ is a pre-trained predictor, the set $\widehat{C}(X_{n+1})$ can be rewritten as

$$[\widehat{\mu}(X_{n+1}) - Q(\widehat{q}; \widehat{F}_{V|X}(\cdot \mid X_{n+1})),$$
$$\widehat{\mu}(X_{n+1}) + Q(\widehat{q}; \widehat{F}_{V|X}(\cdot \mid X_{n+1}))], \quad (3)$$

where $Q(\widehat{q}; \widehat{F}_{V|X}(\cdot \mid X_{n+1}))$ is the $\widehat{q}$ quantile of $\widehat{F}_{V|X}(\cdot \mid X_{n+1})$. In this case, the prediction set at $X_{n+1} = x$ has size $2Q(\widehat{q}; \widehat{F}_{V|X}(\cdot \mid x))$.

Let $\mathcal{D} = (\mathcal{L}_n, \mathcal{D}_{\text{aux}})$, where $\mathcal{D}_{\text{aux}}$ denotes any auxiliary randomness used by the score construction (e.g., pre-trained models or external datasets). Then the law of total variance yields the following decomposition of $\text{Var}(L(\widehat{C}))$ for a general conformal prediction set $\widehat{C}(\cdot)$:

$$\text{Var}(\mathbb{E}\{L(\widehat{C}) \mid \mathcal{D}_{\text{aux}}\}) + \mathbb{E}\{\text{Var}(L(\widehat{C}) \mid \mathcal{D}_{\text{aux}})\}. \quad (4)$$

The first term in (4) measures variation induced by auxiliary components in the score construction. Since the $\mathcal{D}_{\text{aux}}$ determines the score construction of the conformal set and may contain localized information, we keep it unchanged. Therefore, we focus solely on the second term. Throughout the subsequent discussion, we treat $\mathcal{D}_{\text{aux}}$ as fixed. This renders the second term equivalent to setting $\mathcal{D} = \mathcal{L}_n$ in (1), implying that we focus exclusively on the **calibration-sample-induced variation**. By Lemma 4.4 in Section 4, the second term in (4) is controlled by the variability of the score quantile estimator. Let $F_S(s)$ be the marginal CDF of $S(X, Y)$ at $S = s$ for $(X, Y) \sim P$ and its empirical version based on $\mathcal{L}_n$ is denoted by

$$\widehat{F}_S^0(s) = n^{-1} \sum_{i=1}^{n} \mathbb{I}(S(X_i, Y_i) \le s).$$

The $\widehat{q}$ is thus the $1 - \alpha_n$ quantile of $\widehat{F}_S^0(s)$ with $\alpha_n = 1 - (1-\alpha)(n+1)/n$. This implies that the stability of $\widehat{q}$ depends on the stability of $\widehat{F}_S^0(s)$. Therefore, we propose to stabilize conformal prediction by leveraging abundant labeled source data $\mathcal{L}'_N$ and unlabeled target data $\mathcal{U}_m$ to obtain more stable score-distribution and quantile estimation.

## 3. Methodology

Our goal is to improve the stability of conformal prediction sets while retaining satisfactory marginal coverage. The key objective is to estimate $F_S(s)$ more reliably than $\widehat{F}_S^0(s)$ by leveraging the unlabeled data $\mathcal{U}_m$. Write $F_S(s)$ as

$$F_S(s) = \mathbb{E}\left[\mathbb{E}\{\mathbb{I}(S(X, Y) \le s) \mid X\}\right] \qquad (5)$$
$$= \mathbb{E}\left[F_{S|X}(s \mid X)\right], \qquad (6)$$

where the outer expectation is over $X \sim P_X$. This representation shows that estimating $F_S(s)$ from $\mathcal{U}_m$ requires an estimator of the conditional distribution $F_{S|X}$.

We therefore start by estimating $F_{S|X}$. As indicated in (4) and (6), obtaining a stable estimator of $F_{S|X}$ is crucial. In practice, the labeled source data $\mathcal{L}'_N$ are often substantially more abundant than the target labeled data. Although substantial covariate shift may exist between the source and target domains, using $\mathcal{L}'_N$ can potentially yield a more accurate estimator of $F_{S|X}$ under moderate concept shift. Let $\mathcal{A}(\cdot)$ denote a generic algorithm for conditional CDF estimation (Kneib et al., 2023). Using the source data $\mathcal{L}'_N$, we obtain $\widehat{F}_{S|X} = \mathcal{A}(\mathcal{L}'_N)$ as an initial estimator of $F_{S|X}$.

*Remark* 3.1. Due to the distribution shift between $P$ and $P'$, directly merging the source data into the target set and applying the conformal prediction procedure on the combined dataset can severely compromise the marginal validity of the resulting conformal prediction set.

Because $\mathcal{U}_m$ contains a large number of unlabeled samples, the outer expectation in (6) can be approximated using $\mathcal{U}_m$. Accordingly, for any estimator $\widetilde{F}$ of the conditional distribution $S \mid X$, (6) leads to an estimator of $F_S(s)$ as

$$\widehat{F}^1_S(s; \widetilde{F}) = m^{-1} \sum_{j=1}^m \widetilde{F}(s \mid \widetilde{X}_j).$$

A direct construction is to take the $1 - \alpha_n$ quantile of $\widehat{F}^1_S(\cdot; \widetilde{F})$, denoted by $\widehat{q}_{\mathrm{DP}} = \inf \left\{ s : \widehat{F}^1_S(s; \widetilde{F}) \geq 1 - \alpha_n \right\}$, and define the *Direct Plug-in Set (DPS)*

$$\widehat{C}_{\mathrm{DP}}(X_{n+1}) = \{ y : S(X_{n+1}, y) \leq \widehat{q}_{\mathrm{DP}} \}.$$

However, DPS can seriously violate marginal coverage validity, because $\widetilde{F}$ is only an estimated conditional distribution and its misspecification error can be directly translated into bias in $\widehat{F}^1_S$ and thus in $\widehat{q}_{\mathrm{DP}}$.

To reduce this bias, Wen et al. (2025) and prediction-powered correction ideas (Angelopoulos et al., 2023) consider debiased CDF estimators of the form $\breve{F}_S(s) = \widehat{F}^0_S(s) + \widehat{F}^1_S(s; \widetilde{F}) - n^{-1} \sum_{j=1}^n \widetilde{F}(s \mid X_j)$. When $\widetilde{F}$ has sufficient conditional accuracy, this correction can reduce the estimation variance of the marginal CDF. Yet it still does not provide a strict marginal guarantee in general, nor a uniform prediction-set variance-reduction guarantee under weak assumptions.

Motivated by this limitation, we design a transductive calibration strategy that directly controls marginal information through $\widehat{F}^0_S$ while transferring stable information from source data via $\widehat{F}_{S|X}$. We achieve this by recalibrating $\widehat{F}_{S|X}$ with the use of both $\mathcal{U}_m$ and $\mathcal{L}_n$.

Intuitively, with $\widetilde{F}$, the resulting estimator $\widehat{F}^1_S(s; \widetilde{F})$ should not deviate significantly from $\widehat{F}^0_S(s)$ to serve as a good esti-

mator of $F_S(s)$. Thus, we consider calibrating $\widehat{F}_{S|X}$ using $\mathcal{L}_n$ to obtain a refined estimator $\widetilde{F}_{S|X}$. Suppose $\widetilde{F}$ comes from a solution space $\mathcal{F}$ containing $\widehat{F}_{S|X}$. Let $d(\cdot, \cdot)$ be a distance measure between two distributions on $\mathbb{R}$ (e.g., KL divergence or Wasserstein distance), and let $R(\widetilde{F}, \widehat{F}_{S|X})$ be a regularization term penalizing deviations from the initial estimator $\widehat{F}_{S|X}$, with $R(\widehat{F}_{S|X}, \widehat{F}_{S|X}) = 0$. For a tuning parameter $\lambda > 0$, we define the calibrated estimator as

$$\widetilde{F}_{S|X} = \underset{\widetilde{F} \in \mathcal{F}}{\arg\min} \, d\left( \widehat{F}^0_S(\cdot), \widehat{F}^1_S(\cdot; \widetilde{F}) \right) + \lambda R(\widetilde{F}, \widehat{F}_{S|X}). \tag{7}$$

Setting $\lambda = 0$ encourages full alignment between distribution estimate $\widehat{F}^1_S(\cdot; \widetilde{F}_{S|X})$ and $\widehat{F}^0_S(\cdot)$, which uses only $\mathcal{L}_n$. In our implementation, this alignment is carried out on a finite quantile grid that contains $1 - \alpha_n$, so the target conformal quantile is recovered at $\lambda = 0$; see Appendix C.1. In contrast, as $\lambda \to \infty$, $\widetilde{F}_{S|X}$ converges to $\widehat{F}_{S|X}$, thus fully trusting the source-derived estimator. The estimator $\widehat{F}_{S|X}$ serves as a natural starting point for aligning $\widetilde{F}_{S|X}$, since it is estimated from abundant source data and provides a stable and relatively reliable approximation. Adjusting $\lambda$ thus enables smooth interpolation between the purely target-calibrated and the fully source-informed regimes, allowing us to control the weight placed on $\widehat{F}_{S|X}$ and thereby enhance the estimation of $F_S(s)$ on the target data.

Finally, given $\widetilde{F}_{S|X}$, we estimate $Q(1 - \alpha_n; F_S)$, the $1 - \alpha_n$ quantile of $F_S(s)$, by the aligned quantile estimator

$$\widehat{q}_{\mathrm{St}} = \inf \left\{ s : \widehat{F}^1_S(s; \widetilde{F}_{S|X}) \geq 1 - \alpha_n \right\}. \tag{8}$$

We construct the Stable Conformal Prediction (StCP) set via transduction as

$$\widehat{C}_{\mathrm{St}}(X_{n+1}) = \{ y : S(X_{n+1}, y) \leq \widehat{q}_{\mathrm{St}} \}. \tag{9}$$

The StCP procedure for the target task is outlined in Algorithm 1. In the theoretical analysis in Section 4, we demonstrate that the design of StCP ensures robustness against inaccuracies in the initial conditional CDF estimator. Under varying levels of accuracy in the initial estimator, we establish a trade-off between achieving marginal validity and obtaining enhanced stability.

The overall workflow is summarized in Figure 2; in particular, solid boxes represent labeled data, while dashed boxes denote unlabeled data; blue elements correspond to operations on the target data, orange elements to operations on the source data, and green elements highlight the alignment step and the final construction of the StCP set.

Importantly, StCP is not restricted to standard conformal scores. It can be directly applied to conformal methods with localization by replacing the generic score $S$ with the corresponding localized score, leading to Stable Localized

**Algorithm 1** Stable Conformal Prediction (StCP) via transduction

---

**Input:** labeled target data $\mathcal{L}_n = \{(X_i, Y_i)\}_{i=1}^n$, unlabeled target data $\mathcal{U}_m = \{\widetilde{X}_j\}_{j=1}^m$, labeled source data $\mathcal{L}'_N$, score function $S(x,y)$, conditional CDF learner $\mathcal{A}$, candidate space $\mathcal{F}$, coverage level $1-\alpha$, tuning parameter $\lambda$, test point $X_{n+1}$

1. Train an initial conditional CDF estimator from source data: $\widehat{F}_{S|X}(\cdot \mid x) = \mathcal{A}(\mathcal{L}'_N)$
2. Compute scores $S_i = S(X_i, Y_i)$ for $i = 1, \ldots, n$, and form $\widehat{F}_S^0(s) = n^{-1} \sum_{i=1}^n \mathbb{1}(S_i \leq s)$
3. For each $\widetilde{F} \in \mathcal{F}$, compute the transductive marginal CDF estimate $\widehat{F}_S^1(s; \widetilde{F}) = m^{-1} \sum_{j=1}^m \widetilde{F}(s \mid \widetilde{X}_j)$
4. Calibrate the $\widetilde{F}_{S|X}$ by marginal alignment:
$$\arg\min_{\widetilde{F} \in \mathcal{F}} d\left(\widehat{F}_S^0, \widehat{F}_S^1(\cdot; \widetilde{F})\right) + \lambda R(\widetilde{F}, \widehat{F}_{S|X})$$
5. Set $\alpha_n = 1 - (1-\alpha)(n+1)/n$ and compute
$$\widehat{q}_{\mathrm{St}} = \inf\left\{s : \widehat{F}_S^1(s; \widetilde{F}_{S|X}) \geq 1 - \alpha_n\right\}$$
6. Construct $\widehat{C}_{\mathrm{St}}(X_{n+1}) = \{y : S(X_{n+1}, y) \leq \widehat{q}_{\mathrm{St}}\}$

**Return:** $\widehat{q}_{\mathrm{St}}$ and $\widehat{C}_{\mathrm{St}}(X_{n+1})$

---

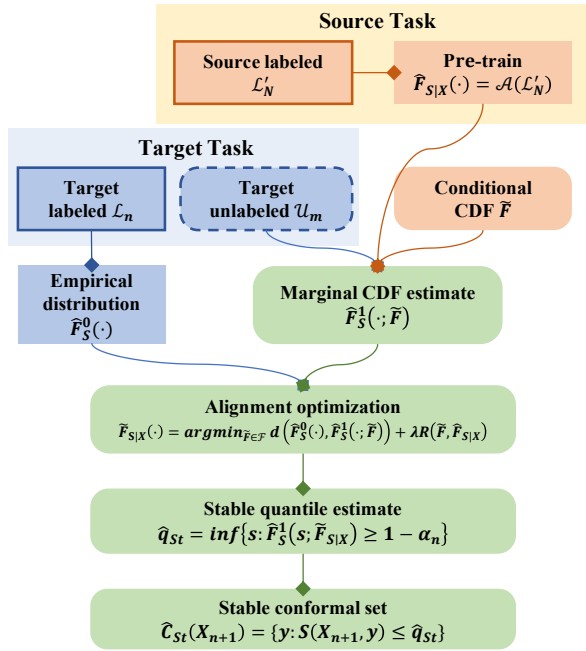

*Figure 2.* Workflow of StCP.

Conformal Prediction (SLCP). The theoretical results in Section 4 continue to apply. In practice, this localized version is especially beneficial: SLCP maintains conditional-coverage performance while achieving substantially larger stability gains, as confirmed by the experiments in Section 5.

### 3.1. Data-driven Parameter Selection of $\lambda$

As shown in Section 5.2, StCP is generally robust to $\lambda$ over a wide range. Nevertheless, when the trustworthiness of the source information is uncertain, a data-driven selection rule is still needed. The detailed pseudocode is provided in Appendix Algorithm 2. Let $\alpha_{\mathrm{tol}} > 0$ be a user-specified tolerance and let $\Lambda$ be a candidate grid of $\lambda$ values. For each $\lambda \in \Lambda$, compute the StCP quantile estimator and denote it by $\widehat{q}_{\mathrm{St},\lambda}$. Define

$$q_{\mathrm{L}} = Q(1 - \alpha - \alpha_{\mathrm{tol}}; \widehat{F}_S^0), \quad q_{\mathrm{U}} = Q(1 - \alpha + \alpha_{\mathrm{tol}}; \widehat{F}_S^0),$$

and the feasible set

$$\Lambda_{\mathrm{feas}}(\alpha, \alpha_{\mathrm{tol}}) = \{\lambda \in \Lambda : \widehat{q}_{\mathrm{St},\lambda} \in [q_{\mathrm{L}}, q_{\mathrm{U}}]\}.$$

We select and define $\widehat{\lambda} = \max \Lambda_{\mathrm{feas}}(\alpha, \alpha_{\mathrm{tol}})$, $\widehat{q}_{\mathrm{St\text{-}sel}} = \widehat{q}_{\mathrm{St},\widehat{\lambda}}$. In our implementation, this rule is well-defined because $\lambda = 0$ belongs to $\Lambda_{\mathrm{feas}}(\alpha, \alpha_{\mathrm{tol}})$: the Wasserstein alignment is computed on a finite quantile grid containing $1 - \alpha_n$, so the resulting StCP quantile at $\lambda = 0$ coincides with the empirical conformal quantile based on $\widehat{F}_S^0$; see Appendix C.1. Since larger $\lambda$ yields stronger variance reduction (Theorem 4.6), selecting the largest feasible value

provides a stability-oriented choice under the tolerance constraint. Finally, define

$$\widehat{C}_{\mathrm{St\text{-}sel}}(X_{n+1}) = \{y : S(X_{n+1}, y) \leq \widehat{q}_{\mathrm{St\text{-}sel}}\}.$$

That is, we construct the final StCP set using the selected parameter $\widehat{\lambda}$.

## 4. Theoretical Analysis

First, we show that StCP preserves satisfactory marginal coverage. When $\lambda = 0$, the optimization problem in (7) reduces to directly aligning $\widehat{F}_S^1(\cdot; \widetilde{F}_{S|X})$ with $\widehat{F}_S^0(\cdot)$; in our implementation, this recovers the standard conformal quantile based on target calibration scores through the finite-grid alignment described in Appendix C.1. When $\lambda \neq 0$, if the source-based initial estimator $\widehat{F}_{S|X}$ is reasonably accurate, StCP remains close to the nominal level $1 - \alpha$.

To formalize this intuition, we establish a bound on the marginal coverage of StCP under a parametric model for the conditional distribution $F_{S|X}$. Suppose the function space $\mathcal{F}$ is parameterized as $\mathcal{F} = \{F(\cdot \mid \cdot; \theta) : \theta \in \Theta\}$, and let $F_{S|X}(\cdot \mid \cdot) = F(\cdot \mid \cdot; \theta^*)$ denote the true conditional score distribution. Assume that the estimator $\widehat{F}_{S|X}(\cdot \mid \cdot)$ takes the form $\widehat{F}_{S|X}(\cdot \mid \cdot) = F(\cdot \mid \cdot; \widehat{\theta})$, where $\theta^*, \widehat{\theta} \in \Theta$. Let $f(\cdot \mid \cdot; \theta)$ denote the density function of $F(\cdot \mid \cdot; \theta)$ for $\theta \in \Theta$. For notational simplicity, we write $F_\theta = F(\cdot \mid \cdot; \theta)$.

We consider the squared $\infty$-Wasserstein distance for $d(\cdot, \cdot)$

and define the regularization term as $R(F_\theta, F_{\widehat{\theta}}) = \|\theta - \widehat{\theta}\|_2^2$ for $F_\theta \in \mathcal{F}$.

**Assumption 4.1.** Suppose that: (i) the parameter space $\Theta$ is bounded under the $\|\cdot\|_2$ norm; (ii) the support of $F(\cdot \mid x; \theta)$ is bounded uniformly in $x$ and $\theta$; (iii) $F(s \mid x; \theta)$ is Lipschitz continuous in $\theta$; and (iv) the density $f(\cdot \mid \cdot; \theta)$ is uniformly bounded away from zero and infinity, and its partial derivatives $\partial f(s \mid x; \theta)/\partial \theta$ and $\partial f(s \mid x; \theta)/\partial s$ are uniformly upper bounded.

**Theorem 4.2.** *Suppose Assumption 4.1 holds. If there exists $\epsilon \geq 0$ such that, for any realization of the data,*

$$\min_{\theta \in \Theta} d\left(\widehat{F}_S^0(\cdot), \widehat{F}_S^1(\cdot; F_\theta)\right) \leq \epsilon^2,$$

*define $\delta_S = \left| Q(1 - \alpha; F_S) - Q\left(1 - \alpha_n; \widehat{F}_S^1(\cdot; F_{\widehat{\theta}})\right) \right|$ for $1 - \alpha_n = (1 - \alpha)(n + 1)/n$. Then there exists a constant $C > 0$ such that:*

$$\left| \mathbb{P}\left(Y_{n+1} \in \widehat{C}_{\mathrm{St}}(X_{n+1})\right) - (1 - \alpha) \right|$$
$$\leq C \min(\epsilon + \lambda^{1/2} + n^{-1}, \delta_S + \lambda^{-1/2} + n^{-1}).$$

*Remark* 4.3. Theorem 4.2 highlights two complementary regimes. If the parameter space $\Theta$ is sufficiently rich so that $\epsilon$ is small, one can choose a small $\lambda$ to keep the adjusted quantile estimator $\widehat{q}_{\mathrm{St}}$ close to the conformal quantile based on $\widehat{F}_S^0$, thereby preserving marginal coverage. Conversely, when $F_{\widehat{\theta}} = \widehat{F}_{S|X}(\cdot \mid \cdot)$ is already a good estimator of $F_{\theta^*} = F_{S|X}(\cdot \mid \cdot)$, $\delta_S$ is small, and a larger $\lambda$ still maintains satisfactory marginal coverage. Overall, Theorem 4.2 formalizes the robustness of StCP to the choice of $\lambda$ under generic score models. Appendix A.3 gives a brief interpretation of the constant $C$ and clarifies how the bound depends on the main regularity parameters.

Next, we turn to the analysis of set stability for conformal sets. The key driver is the variability of the estimated score quantile. Motivated by this observation, we first establish a general relationship between set stability and quantile variance in Lemma 4.4.

**Lemma 4.4.** *Suppose the conditions of Theorem 4.2 hold and the density of $F_S(s)$ is bounded away from zero. Let $\widehat{C}(\cdot)$ be a prediction set of the form $\{y : S(X_{n+1}, y) \leq \widehat{q}\}$, where $\widehat{q}$ is independent of $X_{n+1}$ and $S$ is based on $\mathcal{D}_{\mathrm{aux}}$ independent of $\mathcal{L}_n$. Given $\mathcal{D}_{\mathrm{aux}}$ we assume there exists function $L_0 : [a_0, b_0] \to \mathbb{R}$ such that $L(\widehat{C}) = L_0(\widehat{q})$, and $L_0$ is $C_L$-Lipschitz. Then*

$$\mathrm{Var}(L(\widehat{C}) \mid \mathcal{D}_{\mathrm{aux}}) = O(\mathrm{Var}(\widehat{q} \mid \mathcal{D}_{\mathrm{aux}})).$$

The Lipschitz condition on $L_0$ is mild and is satisfied in standard score constructions where prediction-set size varies smoothly with the score threshold. When the score function is pre-trained and thus independent of the calibration data, the $\mathcal{D}_{\mathrm{aux}}$ can be taken as the pre-training data. Under this setting, the stability of the conformal set is directly controlled by the variance of the quantile estimator $\widehat{q}$.

Accordingly, we take $d(\cdot, \cdot) = W_2^2(\cdot, \cdot)$ due to its differentiability and favorable operational properties. The corresponding extension to the $p$-th power of the $p$-Wasserstein distance for any $p \geq 2$ can be proved similarly, so we focus on the $W_2^2$ case for brevity. Define $R(F_\theta, F_{\widehat{\theta}}) = \|\theta - \widehat{\theta}\|_2^2$ and the population counterpart of $\widehat{F}_S^1$ by $F_S^1(s; F_\theta) = \mathbb{E}\{F(s \mid X; \theta)\}$, where the expectation is taken over $X \sim P_X$.

**Assumption 4.5.** Assume that $\Theta$ is convex, the map $\theta \mapsto d(F_S, F_S^1(\cdot; F_\theta))$ is twice continuously differentiable and there exists a constant $c_d \in \mathbb{R}$ such that $\nabla_{\theta\theta} d(F_S, F_S^1(\cdot; F_\theta)) \succeq c_d I$.

See Appendix A.4 for a brief discussion of this assumption.

**Theorem 4.6.** *Suppose the conditions of Lemma 4.4 and Assumption 4.5 hold. Let $\widehat{C}(X_{n+1}) = \{y : S(X_{n+1}, y) \leq Q(1 - \alpha_n; \widehat{F}_S^0)\}$. Then $\mathrm{Var}(L(\widehat{C}) \mid \mathcal{D}_{\mathrm{aux}}) = O(n^{-1})$. Moreover, if $c_d > 0$, then*

$$\mathrm{Var}(L(\widehat{C}_{\mathrm{St}}) \mid \mathcal{D}_{\mathrm{aux}}) = O\left(m^{-1} + \{n(1 + \lambda)^2\}^{-1}\right).$$

*If $c_d \leq 0$ and $\lambda \geq 1 - c_d$, then*

$$\mathrm{Var}(L(\widehat{C}_{\mathrm{St}}) \mid \mathcal{D}_{\mathrm{aux}}) = O\left(m^{-1} + (n\lambda^2)^{-1}\right).$$

Conditioning on $\mathcal{D}_{\mathrm{aux}}$, these conditional variances in Theorem 4.6 coincide with the notion of set stability.

Theorem 4.6 implies that the standard conformal quantile based on $n$ calibration samples achieves the variance rate $O(n^{-1})$. Nevertheless, when $n$ is small, this can lead to high instability. In contrast, the stability of the StCP set is of order $O(m^{-1} + \{n(1 + \lambda)^2\}^{-1})$ when the population alignment term is already locally convex ($c_d > 0$), and of order $O(m^{-1} + (n\lambda^2)^{-1})$ once $\lambda$ is large enough to dominate a nonpositive lower curvature bound ($c_d \leq 0$ and $\lambda \geq 1 - c_d$). In both regimes, the first term depends on the number of unlabeled samples and is negligible when $m \gg n$, while the second term decreases as the regularization level increases.

The preceding two theorems quantify coverage and stability for a fixed $\lambda$. We next show that the data-driven rule in Section 3.1 also enjoys a finite-sample marginal guarantee, without any parametric assumptions: it only requires exchangeability of the $n$ calibration scores and the test score.

**Theorem 4.7.** *Let $S_i = S(X_i, Y_i)$ be exchangeable for $i = 1, \ldots, n + 1$. We choose $\widehat{\lambda}$ with specified tolerance $\alpha_{\mathrm{tol}} > 0$ as in Section 3.1. Then*

$$\mathbb{P}\left(Y_{n+1} \in \widehat{C}_{\mathrm{St\text{-}sel}}(X_{n+1})\right)$$
$$\in \left[1 - \alpha - \alpha_{\mathrm{tol}}, 1 - \alpha + \alpha_{\mathrm{tol}} + (n + 1)^{-1}\right].$$

*Table 1.* Real data experiment results. In the Marginal block, "$-$" indicates marginal coverage below $1 - \alpha - 0.01$, and "$+$" indicates marginal coverage above $1 - \alpha + 1/(n+1)$. When the same superscript appears in the Std or Size blocks, it indicates that the corresponding method has the same signed marginal-coverage deviation.

| | | | GLCP-type | | | | | | | CQR-type | | | | | | |
|---|---|---|---|---|---|---|---|---|---|---|---|---|---|---|---|---|
| | Dataset | $n/m$ | base | SDCP | PPI | ours | ours-sel | oracle | DP | base | SDCP | PPI | ours | ours-sel | oracle | DP |
| Std | CRIME | 30/500 | 0.75 | 1.42 | 0.78 | **0.50** (42.9%) | **0.58** (27.7%) | 0.16 | 0.86 | 0.55 | 0.53 | 0.53 | **0.42** (25.0%) | **0.44** (19.4%) | 0.10 | 0.86 |
| | BIO | 30/1000 | 0.53 | 0.66 | 0.58 | **0.37** (35.7%) | **0.49** (8.3%) | 0.07 | *0.31+* | 0.42 | 0.39 | 0.41 | **0.32** (29.3%) | **0.36** (12.3%) | 0.14 | *0.31+* |
| | STAR | 30/1000 | 8.78 | 13.18 | 10.42 | **5.25** (48.4%) | **5.65** (42.9%) | 1.48 | *9.13+* | 6.45 | 7.87 | 6.25 | **5.90** (7.3%) | **5.93** (6.7%) | 1.42 | *8.80+* |
| | DERMA | 30/1000 | 0.68 | 0.72 | *0.78+* | **0.49** (30.4%) | **0.46** (35.8%) | 0.06 | *0.12+* | 0.15 | 0.22 | 0.15 | **0.14** (22.1%) | **0.13** (29.3%) | 0.09 | *0.09+* |
| | TISSUE | 30/1000 | 0.83 | 1.23 | 1.02 | **0.73** (13.5%) | **0.79** (4.8%) | 0.12 | *0.07+* | 0.72 | 1.12 | 0.97 | **0.62** (15.4%) | **0.64** (11.5%) | 0.10 | *0.09+* |
| Marginal | CRIME | 30/500 | 0.896 | 0.904 | 0.901 | 0.895 | 0.898 | 0.895 | 0.911 | 0.896 | 0.895 | 0.898 | 0.903 | 0.900 | 0.898 | 0.909 |
| | BIO | 30/1000 | 0.902 | 0.915 | 0.906 | 0.930 | 0.921 | 0.900 | 0.956+ | 0.896 | 0.913 | 0.907 | 0.916 | 0.914 | 0.901 | 0.959+ |
| | STAR | 30/1000 | 0.899 | 0.911 | 0.902 | 0.919 | 0.918 | 0.902 | 0.939+ | 0.892 | 0.905 | 0.899 | 0.895 | 0.895 | 0.896 | 0.935+ |
| | DERMA | 30/1000 | 0.930 | 0.915 | 0.937+ | 0.933 | 0.932 | 0.900 | 0.976+ | 0.925 | 0.925 | 0.931 | 0.924 | 0.930 | 0.901 | 0.964+ |
| | TISSUE | 30/1000 | 0.909 | 0.926 | 0.923 | 0.928 | 0.923 | 0.905 | 0.998+ | 0.905 | 0.917 | 0.920 | 0.925 | 0.920 | 0.905 | 0.994+ |
| Size | CRIME | 30/500 | 3.24 | 3.75 | 3.37 | **3.11** | 3.19 | 3.01 | 3.46 | 3.16 | **3.14** | 3.18 | 3.17 | 3.15 | 3.02 | 3.40 |
| | BIO | 30/1000 | **1.96** | 2.11 | 2.05 | 2.13 | 2.13 | 1.75 | *2.54+* | **2.01** | 2.11 | 2.08 | 2.09 | 2.10 | 1.93 | *2.49+* |
| | STAR | 30/1000 | **39.84** | 44.76 | 41.89 | 41.13 | 41.38 | 37.60 | *47.28+* | **39.05** | 41.40 | 40.60 | 39.12 | 39.11 | 38.41 | *46.16+* |
| | DERMA | 30/1000 | 2.32 | **2.15** | *2.46+* | 2.22 | 2.20 | 1.70 | *3.15+* | 2.66 | 2.69 | 2.67 | **2.65** | 2.65 | 2.60 | *2.71+* |
| | TISSUE | 30/1000 | **4.30** | 4.83 | 4.68 | 4.62 | 4.51 | 4.06 | *7.57+* | **4.22** | 4.58 | 4.59 | 4.57 | 4.49 | 4.10 | *7.35+* |
| Miscoverage | CRIME | 30/500 | 0.040 | 0.035 | 0.039 | 0.043 | 0.039 | 0.040 | 0.042 | 0.043 | 0.045 | 0.042 | 0.038 | 0.039 | 0.040 | 0.043 |
| | BIO | 30/1000 | 0.068 | 0.072 | 0.069 | 0.084 | 0.077 | 0.074 | 0.096 | 0.074 | 0.078 | 0.074 | 0.079 | 0.078 | 0.077 | 0.096 |
| | STAR | 30/1000 | 0.030 | 0.026 | 0.030 | 0.034 | 0.032 | 0.031 | 0.043 | 0.027 | 0.024 | 0.027 | 0.027 | 0.027 | 0.028 | 0.038 |
| | DERMA | 30/1000 | 0.058 | 0.062 | 0.060 | 0.061 | 0.061 | 0.081 | 0.083 | 0.037 | 0.036 | 0.043 | 0.037 | 0.043 | 0.036 | 0.075 |
| | TISSUE | 30/1000 | 0.066 | 0.066 | 0.066 | 0.072 | 0.071 | 0.072 | 0.098 | 0.072 | 0.071 | 0.070 | 0.072 | 0.073 | 0.074 | 0.094 |

## 5. Numerical Experiments

We conduct experiments on five real-world datasets to evaluate the empirical performance of StCP under GLCP-type and CQR-type base methods. The nominal coverage level is fixed at $1 - \alpha = 90\%$ throughout. For each method, we evaluate the prediction set $\widehat{C}(X_{n+1})$ using: (1) marginal coverage $\mathbb{P}(Y_{n+1} \in \widehat{C}(X_{n+1}))$; (2) mean set size $\mathbb{E}\{L(\widehat{C})\}$; (3) test-conditional miscoverage error $\mathbb{E}\{|\mathbb{P}(Y_{n+1} \in \widehat{C}(X_{n+1}) \mid X_{n+1}) - (1 - \alpha)|\}$; and (4) set stability quantified by $\sqrt{\mathrm{Var}(L(\widehat{C}))}$. All experiments are repeated 50 times. The code for all experiments is available at https://github.com/OswinMin/StCP.

### 5.1. Real Data Experiments

We evaluate the proposed method on several publicly available regression and classification datasets that have also been studied in prior conformal prediction work (Romano et al., 2019; Sesia & Romano, 2021; Humbert et al., 2023; Min et al., 2026). These include physicochemical properties of protein tertiary structure (BIO) (Rana,

2013), communities and crimes (CRIME) (Redmond, 2011), Tennessee's Student–Teacher Achievement Ratio dataset (STAR) (Achilles et al., 2008), and two MedMNIST image benchmarks (Yang et al., 2021): DERMA (Dermatoscope) and TISSUE (Kidney Cortex Microscope).

We first split each dataset into target and source subsets to induce a distribution shift; the detailed splitting strategy is described in Section C.3 in the appendix. The values of $n$ and $m$ are those reported in Table 1. For conditional distribution estimation, we use an engression network (Shen & Meinshausen, 2024) with hidden layers $(50, 100, 100, 50)$, and fine-tune only the $100 \times 100$ weight matrix with the number of tunable parameters $k_0 = 10000$. The penalty term takes the quadratic form

$$R(F_\theta, F_{\widehat{\theta}}) = \|\theta - \widehat{\theta}\|_2^2 / k_0,$$

and we search $\lambda$ over $[0, 3000]$. For regression tasks, we use the residual score for $V(x, y)$. For classification tasks, $\widehat{\mu}$ outputs a class-probability vector, and the score is defined as $V(x, y) = 1 - \widehat{\mu}(x)_y$, where $\widehat{\mu}(x)_y$ is the predicted probability of class $y$.

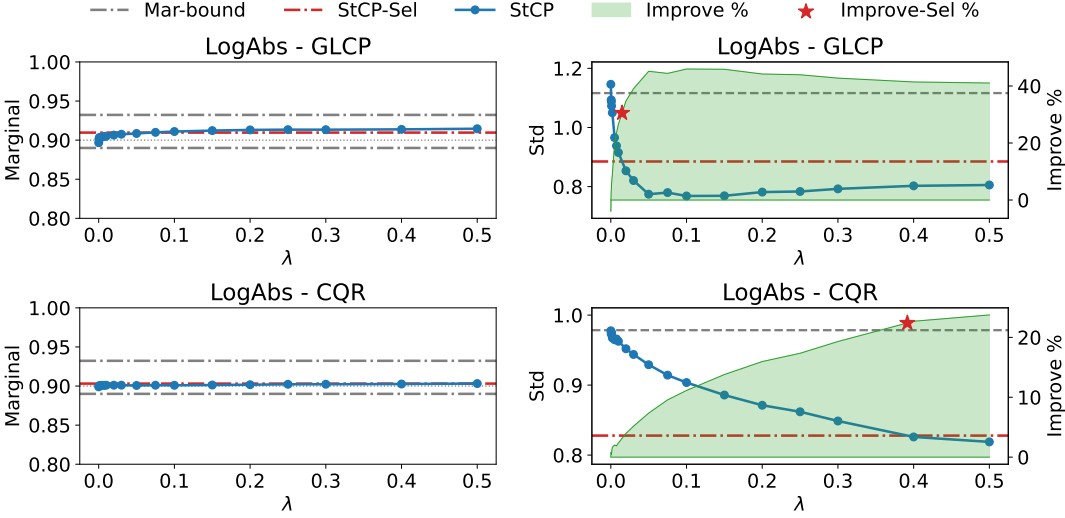

*Figure 3.* Sensitivity to $\lambda$ under the LogAbs setting.

Table 1 reports results for both GLCP-type and CQR-type base methods with the following competitors: *base*, SDCP, PPI, ours, ours-sel, oracle, and DP. Here SDCP denotes the method of Wen et al. (2025), which estimates the conditional distribution of $S \mid X$ using $\mathcal{L}_n$ and then constructs a debiased estimator of the marginal distribution of $S$ in a PPI-style manner. PPI denotes the counterpart that estimates $S \mid X$ from source data $\mathcal{L}'_N$ and then uses a PPI-style debiased marginal estimator. "ours" reports the best Std result over the grid of $\lambda$ while maintaining marginal coverage, whereas "ours-sel" uses the data-driven rule in Section 3.1 with $\alpha_{\text{tol}} = 0.02$. "oracle" labels unlabeled target data and merges them into calibration before applying the corresponding base method. "DP" denotes the direct plug-in variant based on $\widehat{F}_S^1$ without correction.

Our primary objective is to reduce standard deviation while preserving marginal validity. Across both GLCP-type and CQR-type settings, ours consistently delivers lower standard deviation than base, SDCP, and PPI, and ours-sel preserves this improvement pattern while enforcing a more conservative, data-driven choice of $\lambda$. In the Marginal block, ours and ours-sel remain mostly within the acceptable range and avoid the severe upward deviations frequently seen for DP.

Beyond stability, size and miscoverage of ours and ours-sel are generally comparable to those of the corresponding base methods. In a few cases, size is slightly larger for ours, which is consistent with its marginal coverage being slightly higher in those settings; this reflects a coverage–efficiency tradeoff rather than instability. Oracle remains the strongest reference because it uses true labels for the unlabeled target data, and the remaining gap to oracle quantifies room for further improvement under imperfect transfer.

## 5.2. Sensitivity to Sample Size and Parameter $\lambda$

We next study how StCP responds to the tuning parameter $\lambda$ and to the amount of labeled calibration data in a synthetic transfer-regression setting. Let $d = 5$ and write $\mathbf{1}_d$ for the $d$-dimensional all-one vector. We set $\mu_s = 0$ and $\mu_t = \mathbf{1}_d/2\sqrt{d}$, and generate the source and target covariates as $X' \sim N(\mu_s, I_d), X \sim N(\mu_t, I_d)$ and corresponding labels as $Y' = \frac{3}{d}\sum_{j=1}^d X'_j + \varepsilon', Y = \frac{2}{d}\sum_{j=1}^d X_j + \varepsilon$, with the noise terms $\varepsilon'$ and $\varepsilon$ being heteroscedastic and dependent on the covariates:

$$\varepsilon' \mid X' \sim N\big(0, \sigma^2(X'; \gamma_s)\big), \varepsilon \mid X \sim N\big(0, \sigma^2(X; \gamma_t)\big)$$

where $\sigma(x; \gamma) = \sqrt{\gamma/d}\sum_{j=1}^d \log(1 + |x_j|)$ and $\gamma_s = 1.2, \gamma_t = 1$. Thus the synthetic study contains both covariate shift and heteroscedastic label noise, while keeping the target setting analytically transparent. We refer to this setting as 'LogAbs', with $(n, m) = (30, 500)$ for the $\lambda$ study and $m = 500$ with $n \in \{30, 100, 500\}$ for the sample-size study; the additional 'Quad' and 'Softplus' settings are reported in Appendix C.2.

Figure 3 shows that the effect of $\lambda$ is stable and interpretable in this regime. For the GLCP-type base method, increasing $\lambda$ from very small values yields a visible drop in the Std metric while keeping marginal coverage near the nominal level over a broad interval. For the CQR-type variant, the improvement is milder but still persistent, which is consistent with the smaller stability gains already observed in the real-data study. The curves also flatten as $\lambda$ becomes large, indicating that the benefit from stronger source regularization eventually saturates rather than increasing indefinitely.

Table 2 examines the effect of the labeled calibration size $n$. The gain is strongest when $n = 30$, where calibration

*Table 2.* Simulation results under the LogAbs setting with $m = 500$ and $n \in \{30, 100, 500\}$. In the Marginal block, "$-$" indicates marginal coverage below $1 - \alpha - 0.01$, and "$+$" indicates marginal coverage above $1 - \alpha + 1/(n+1)$.

| | Setting | $n$ | Model | base | SDCP | PPI | ours | ours-sel | oracle | DP |
|---|---|---|---|---|---|---|---|---|---|---|
| | | | | | | | | | | Method |
| Std | LogAbs | 30 | GLCP | 1.12 | $1.65^+$ | 1.09 | **0.77** (31.2%) | **0.88** (20.7%) | 0.36 | 0.85 |
| | | | CQR | 0.98 | 0.88 | 0.94 | **0.82** (16.3%) | **0.83** (15.4%) | 0.31 | 0.81 |
| | LogAbs | 100 | GLCP | 0.50 | $0.67^-$ | 0.46 | **0.38** (23.2%) | $0.38^+$ (24.4%) | 0.15 | $0.41^+$ |
| | | | CQR | 0.45 | $0.46^-$ | 0.47 | **0.38** (16.7%) | **0.38** (15.8%) | 0.14 | $0.38^+$ |
| | LogAbs | 500 | GLCP | 0.25 | $0.21^-$ | 0.21 | **0.18** (26.6%) | $0.16^+$ (33.9%) | 0.14 | $0.21^+$ |
| | | | CQR | 0.18 | $0.17^-$ | 0.18 | **0.17** (6.3%) | **0.17** (6.3%) | 0.13 | $0.20^+$ |
| Marginal | LogAbs | 30 | GLCP | 0.899 | $0.935^+$ | 0.909 | 0.911 | 0.910 | 0.904 | 0.921 |
| | | | CQR | 0.899 | 0.896 | 0.905 | 0.903 | 0.903 | 0.899 | 0.916 |
| | LogAbs | 100 | GLCP | 0.902 | $0.867^-$ | 0.908 | 0.910 | $0.910^+$ | 0.902 | $0.924^+$ |
| | | | CQR | 0.899 | $0.884^-$ | 0.902 | 0.900 | 0.900 | 0.900 | $0.920^+$ |
| | LogAbs | 500 | GLCP | 0.900 | $0.873^-$ | 0.900 | 0.902 | $0.909^+$ | 0.902 | $0.920^+$ |
| | | | CQR | 0.900 | $0.885^-$ | 0.901 | 0.899 | 0.899 | 0.900 | $0.918^+$ |
| Size | LogAbs | 30 | GLCP | **5.19** | $6.20^+$ | 5.39 | 5.36 | 5.34 | 5.04 | 5.65 |
| | | | CQR | 5.24 | **5.14** | 5.32 | 5.24 | 5.24 | 5.03 | 5.48 |
| | LogAbs | 100 | GLCP | **4.81** | $4.35^-$ | 4.93 | 4.92 | $4.93^+$ | 4.74 | $5.23^+$ |
| | | | CQR | **4.84** | $4.63^-$ | 4.88 | **4.82** | **4.83** | 4.79 | $5.15^+$ |
| | LogAbs | 500 | GLCP | **4.63** | $4.24^-$ | 4.64 | 4.65 | $4.78^+$ | 4.64 | $5.01^+$ |
| | | | CQR | **4.71** | $4.52^-$ | 4.72 | **4.69** | **4.69** | 4.71 | $4.98^+$ |
| Miscoverage | LogAbs | 30 | GLCP | 0.014 | 0.035 | 0.017 | 0.017 | 0.016 | 0.015 | 0.023 |
| | | | CQR | 0.019 | 0.020 | 0.019 | 0.019 | 0.019 | 0.020 | 0.021 |
| | LogAbs | 100 | GLCP | 0.014 | 0.035 | 0.017 | 0.016 | 0.016 | 0.014 | 0.026 |
| | | | CQR | 0.021 | 0.026 | 0.020 | 0.021 | 0.020 | 0.021 | 0.025 |
| | LogAbs | 500 | GLCP | 0.015 | 0.029 | 0.016 | 0.015 | 0.016 | 0.014 | 0.023 |
| | | | CQR | 0.021 | 0.026 | 0.021 | 0.021 | 0.021 | 0.021 | 0.024 |

noise is most severe: under GLCP, the Std metric decreases from 1.12 to 0.77, and under CQR it decreases from 0.98 to 0.82, while the marginal coverage remains within an acceptable range. As $n$ increases, the room for variance reduction naturally becomes smaller because the baseline conformal quantile itself becomes more stable. Even so, StCP remains competitive at $n = 500$, suggesting that the method is particularly valuable in low-label regimes without introducing an obvious downside once more calibration labels are available.

Also, in this experiment a range of marginal coverage values around the nominal level remains acceptable, but even within that acceptable interval the prediction-set size is still noticeably affected by how conservative the method is. Consequently, in some cases ours or ours-sel has a slightly larger size than base simply because it attains a slightly larger marginal coverage. The CQR results illustrate this clearly: when $n = 30$, ours and ours-sel achieve essentially the same size as base (5.24) while attaining a slightly larger marginal coverage (0.903 versus 0.899); when $n = 100$, they attain a slightly larger marginal coverage (0.900 versus 0.899) together with a slightly smaller size (4.82/4.83 versus 4.84). Meanwhile, the miscoverage values remain fully comparable to those of base and are nearly identical in these two settings, so the observed size differences are best under-

stood as reflecting small shifts in marginal conservativeness rather than a meaningful deterioration in predictive quality.

## 6. Conclusion

This paper introduces *set stability* as a reliability criterion for addressing the instability of conformal prediction with limited calibration data. We propose Stable Conformal Prediction (StCP), a transfer learning approach that stabilizes prediction sets by leveraging labeled source data together with unlabeled target data. StCP can be seamlessly integrated with localized methods, yielding Stable Localized Conformal Prediction (SLCP). Theoretically, StCP preserves valid coverage while reducing variability in set size; empirically, it produces more concentrated prediction sets.

## Impact Statement

This paper presents work whose goal is to advance the field of machine learning. There are many potential societal consequences of our work, none of which we feel must be specifically highlighted here.

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

# A. Details and Verifications of Assumptions

## A.1. Data-driven Parameter Selection Algorithm

Here we provide the corresponding algorithm for the data-driven parameter selection described in Section 3.1.

---

**Algorithm 2** Stable conformal prediction with data-driven parameter selection

---

**Input:** calibration scores $\{S_i\}_{i=1}^n$, unlabeled data $\mathcal{U}_m$, candidate grid $\Lambda$, tolerance $\alpha_{\text{tol}}$, and nominal level $1 - \alpha$.

1. Compute $\widehat{F}_S^0(s) = n^{-1} \sum_{i=1}^n \mathbb{I}(S_i \leq s)$.

2. Set $q_{\text{L}} = Q(1 - \alpha - \alpha_{\text{tol}}; \widehat{F}_S^0)$ and $q_{\text{U}} = Q(1 - \alpha + \alpha_{\text{tol}}; \widehat{F}_S^0)$.

3. For each $\lambda \in \Lambda$, compute $\widehat{q}_{\text{St},\lambda}$ by the StCP procedure with tuning parameter $\lambda$.

4. Form $\Lambda_{\text{feas}} = \{\lambda \in \Lambda : \widehat{q}_{\text{St},\lambda} \in [q_{\text{L}}, q_{\text{U}}]\}$.

5. Select $\widehat{\lambda} = \max \Lambda_{\text{feas}}$ and set $\widehat{q}_{\text{St-sel}} = \widehat{q}_{\text{St},\widehat{\lambda}}$.

6. Construct $\widehat{C}_{\text{St-sel}}(X_{n+1}) = \{y : S(X_{n+1}, y) \leq \widehat{q}_{\text{St-sel}}\}$.

**Return:** $\widehat{C}_{\text{St-sel}}$.

---

## A.2. Validation of Assumption 4.1

Here we provide a brief justification for Assumption 4.1 under standard regularity conditions used in parametric conditional distribution modeling.

**(i) Bounded parameter space.** In practice, optimization is carried out over a bounded region (e.g., by constrained training, explicit regularization, or bounded initialization neighborhoods), so it is natural to work with a compact parameter set $\Theta$.

**(ii) Uniformly bounded support.** Many score constructions in conformal prediction are based on transformed residual-type quantities with a practically truncated range, and distributional models are often built on compact domains for both theory and implementation. Under such settings, the support of $F(\cdot \mid x; \theta)$ is uniformly bounded in $(x, \theta)$.

**(iii) Lipschitz continuity in $\theta$.** If $F(s \mid x; \theta)$ is continuously differentiable in $\theta$ and $\Theta$ is compact, then

$$\sup_{s,x,\theta} \|\partial_\theta F(s \mid x; \theta)\|_2 < \infty,$$

which implies a global Lipschitz bound in $\theta$ by the mean-value theorem. This is standard for smooth parametric families and neural-network parameterizations with bounded weights and Lipschitz activations. This property underlies the stability analysis in Section 3 of Hardt et al. (2016), which relies on the loss (and thus the model output) being Lipschitz in the parameters.

**(iv) Density bounded away from zero and infinity and bounded first derivatives.** Assume $f(s \mid x; \theta)$ is continuous in $(s, x, \theta)$ on a compact domain and strictly positive. Then compactness yields finite constants

$$0 < \underline{L}_F \leq f(s \mid x; \theta) \leq \overline{L}_F < \infty$$

uniformly. If, in addition, $f$ is continuously differentiable in $(s, \theta)$ on the same compact domain, then $\partial_\theta f$ and $\partial_s f$ are uniformly bounded. These conditions are standard in localized or distributional conformal analyses; see, e.g., Guan (2023, Assumption 1).

Overall, Assumption 4.1 is satisfied by a broad class of smooth parametric conditional distribution estimators under compact-domain modeling and routine boundedness regularization.

## A.3. Interpretation of the constant in the marginal coverage bound

The constant $C$ in Theorem 4.2 collects the regularity constants that appear when translating perturbations of the conditional score distribution into perturbations of the target quantile and hence into marginal coverage error. Under Assumption 4.1, it

can be taken proportional to three main ingredients: (i) a norm bound on the parameter space $\Theta$, (ii) the reciprocal of the lower bound of the conditional density $f(s \mid x; \theta)$, and (iii) the Lipschitz constant of $F(s \mid x; \theta)$ with respect to $\theta$.

A larger parameter-space bound or a larger Lipschitz constant means that small perturbations in the fitted score model can induce larger changes in the conditional distribution estimate, which in turn amplifies the deviation of the aligned quantile estimator. Meanwhile, when the density lower bound is very small, the quantile map becomes ill-conditioned: even a small perturbation in the marginal CDF may lead to a comparatively large shift in the corresponding quantile. Therefore, the coverage deviation bound in Theorem 4.2 becomes looser in poorly conditioned regimes and tighter when the model class is well regularized and the score density near the target quantile is bounded away from zero.

### A.4. Validation of Assumption 4.5

We briefly discuss why Assumption 4.5 is natural in smooth finite-dimensional parametric models.

**Convexity and curvature.** The convexity of $\Theta$ is a standard technical device ensuring that the line segment joining $\theta_\lambda$ and $\widetilde{\theta}$ remains inside the parameter space, so the mean-value expansion in the proof of Theorem 4.6 is legitimate. The lower Hessian bound

$$\nabla_{\theta\theta} d\big(F_S, F_S^1(\cdot; F_\theta)\big) \succeq c_d I$$

is exactly the population curvature requirement used to control the inverse Hessian of $J_\lambda$ after adding the quadratic regularizer.

**Pointwise Hessian consistency.** Although it is no longer stated as part of Assumption 4.5, the proof uses the pointwise relation

$$\left\|\nabla_{\theta\theta}\widehat{J}_\lambda(\theta_\lambda) - \nabla_{\theta\theta} J_\lambda(\theta_\lambda)\right\|_{\mathrm{op}} = o_p(1).$$

For $d(\cdot, \cdot) = W_2^2(\cdot, \cdot)$,

$$\nabla_{\theta\theta}\widehat{J}_\lambda(\theta) = 2\int_0^1 \left[\nabla_\theta\widehat{q}^1(u;\theta)\nabla_\theta\widehat{q}^1(u;\theta)^\top + \{\widehat{q}^1(u;\theta) - \widehat{q}^0(u)\}\nabla_{\theta\theta}\widehat{q}^1(u;\theta)\right] du + 2\lambda I,$$

with the analogous population formula for $J_\lambda$. At the fixed optimizer $\theta_\lambda$, each Hessian entry is therefore a finite sum of integrals involving pointwise differences of $\widehat{q}^0 - q^0$, $\widehat{q}^1 - q^1$, $\nabla_\theta\widehat{q}^1 - \nabla_\theta q^1$, and $\nabla_{\theta\theta}\widehat{q}^1 - \nabla_{\theta\theta}q^1$. The first three terms are handled exactly as in the main proof. The last term follows by differentiating the quantile identity $F_S^1(q^1(u;\theta); F_\theta) = u$ once more: under the same local smoothness that gives bounded $\partial_\theta F$, $\partial_{\theta\theta}F$, $\partial_\theta f$, and $\partial_s f$, the second derivative $\nabla_{\theta\theta}\widehat{q}^1(u;\theta_\lambda)$ is again a ratio of empirical averages of bounded quantities, so ordinary pointwise laws of large numbers imply entrywise $o_p(1)$. Since the parameter dimension is fixed, entrywise convergence implies the operator-norm statement above. This remains a fixed-point argument and does not require any uniform control over the whole parameter space.

**When these local conditions hold.** The pointwise Hessian argument above is valid, for example, if $F(s \mid x; \theta)$ is twice continuously differentiable in $\theta$ and continuously differentiable in $s$ on a compact neighborhood of $\theta_\lambda$, with the relevant derivatives locally Lipschitz and uniformly bounded. In particular, bounded $\partial_\theta F$, $\partial_{\theta\theta}F$, $\partial_\theta f$, and $\partial_s f$ are enough to justify the pointwise Hessian consistency discussed above. This covers the usual smooth generalized linear, location-scale, and bounded-weight neural-network distributional models used in conformal prediction.

## B. Additional Lemmas and Proof of Lemmas and Theorems

### B.1. Preliminary Lemmas

**Lemma B.1.** *For any CDFs $F_1(s)$ and $F_2(s)$ both supported on $[a, b]$, if $\sup_{s\in[a,b]} |F_1(s) - F_2(s)| \leq \varepsilon$, and there exists $i \in \{1, 2\}$ such that the density of $F_i(s)$ is lower bounded by $\underline{L}_F > 0$, then for any $\alpha \in (0, 1)$, we have*

$$|Q(1 - \alpha; F_1) - Q(1 - \alpha; F_2)| \leq \varepsilon/\underline{L}_F.$$

### B.2. Proof of Lemma B.1

*Proof.* Let $u = 1 - \alpha \in (0, 1)$, and define the left-continuous quantiles

$$q_1 = Q(u; F_1) = \inf\{s : F_1(s) \geq u\}, \qquad q_2 = Q(u; F_2) = \inf\{s : F_2(s) \geq u\}.$$

Assume first that $F_1$ has density $f_1$ with $f_1(s) \geq \underline{L}_F > 0$ on $[a, b]$. Then $F_1$ is strictly increasing, and for any $x < y$,

$$F_1(y) - F_1(x) = \int_x^y f_1(t)\,dt \geq \underline{L}_F(y - x).$$

Hence its inverse quantile map is $(1/\underline{L}_F)$-Lipschitz:

$$|Q(v_1; F_1) - Q(v_2; F_1)| \leq |v_1 - v_2|/\underline{L}_F, \qquad v_1, v_2 \in (0, 1).$$

Since $\sup_s |F_1(s) - F_2(s)| \leq \varepsilon$, for any $s$ we have

$$F_1(s) \leq u - \varepsilon \Rightarrow F_2(s) < u, \qquad F_1(s) \geq u + \varepsilon \Rightarrow F_2(s) \geq u.$$

By the definition of $q_2 = \inf\{s : F_2(s) \geq u\}$, this implies

$$Q(u - \varepsilon; F_1) \leq q_2 \leq Q(u + \varepsilon; F_1),$$

where boundary values are understood with clipping to $[0, 1]$ if needed. Therefore,

$$|q_2 - q_1| \leq \max\{|Q(u + \varepsilon; F_1) - Q(u; F_1)|,\ |Q(u; F_1) - Q(u - \varepsilon; F_1)|\} \leq \varepsilon/\underline{L}_F.$$

If instead $F_2$ has density lower bounded by $\underline{L}_F$, the same argument with the roles of $(F_1, q_1)$ and $(F_2, q_2)$ exchanged yields the identical bound. Thus

$$|Q(1 - \alpha; F_1) - Q(1 - \alpha; F_2)| \leq \varepsilon/\underline{L}_F.$$

$$\square$$

### B.3. Proof of Theorem 4.2

*Proof.* For notation simplicity, for any $\theta \in \Theta$ define

$$F_\theta = F(s \mid x; \theta), \qquad \widehat{F}_S^1(s; F_\theta) = m^{-1} \sum_{j=1}^m F(s \mid \widetilde{X}_j; \theta).$$

Denote the quantiles

$$\widehat{q}_0 = Q\left(1 - \alpha_n; \widehat{F}_S^1(\cdot; F_{\widehat{\theta}})\right), \quad \widehat{q} = Q\left(1 - \alpha_n; \widehat{F}_S^0\right), \quad \widehat{q}_{\text{St}} = Q\left(1 - \alpha_n; \widehat{F}_S^1(\cdot; F_{\widetilde{\theta}})\right),$$

and $\widehat{Q}(\theta) = Q\left(1 - \alpha_n; \widehat{F}_S^1(\cdot; F_\theta)\right)$. Since $\widehat{q}_{\text{St}}$ is independent of $(X_{n+1}, Y_{n+1})$,

$$\mathbb{P}\left(Y_{n+1} \in \widehat{C}_{\text{St}}(X_{n+1})\right) = \mathbb{E}\{F_S(\widehat{q}_{\text{St}})\}.$$

By Assumption 4.1(iii), there exists $L_\theta > 0$ such that

$$|F(s \mid x; \theta_1) - F(s \mid x; \theta_2)| \leq L_\theta \|\theta_1 - \theta_2\|_2,$$

hence

$$\left|\widehat{F}_S^1(s; F_{\theta_1}) - \widehat{F}_S^1(s; F_{\theta_2})\right| \leq m^{-1} \sum_{j=1}^m \left|F(s \mid \widetilde{X}_j; \theta_1) - F(s \mid \widetilde{X}_j; \theta_2)\right|$$

$$\leq L_\theta \|\theta_1 - \theta_2\|_2.$$

By Assumption 4.1(iv), the density of $F(s \mid x; \theta)$ is lower bounded by $\underline{L}_F > 0$. Since $\widehat{F}_S^1(\cdot; F_\theta)$ is a linear combination of $F(s \mid \widetilde{X}_j; \theta)$, its density is also lower bounded by $\underline{L}_F > 0$. By Lemma B.1, $\widehat{Q}(\theta)$ is Lipschitz in $\theta$ with constant $L_\Theta = L_\theta/\underline{L}_F$.

Also by Assumption 4.1(iv), the density of $F(s \mid x; \theta)$ is upper bounded by $\overline{L}_F < \infty$. Since $F_S(s) = \mathbb{E}\{F(s \mid X; \theta^*)\}$ is a linear integral of $F(s \mid X; \theta^*)$, its density is also upper bounded by $\overline{L}_F < \infty$. Let $\overline{L}_F$ be the Lipschitz constant of $F_S$. Then

$$|\mathbb{E}\{F_S(\widehat{q}_{\mathrm{St}})\} - \mathbb{E}\{F_S(\widehat{q}_0)\}| \le \overline{L}_F \mathbb{E}(|\widehat{q}_{\mathrm{St}} - \widehat{q}_0|) \le L_\Theta \overline{L}_F \mathbb{E}(\|\widetilde{\theta} - \widehat{\theta}\|_2), \tag{10}$$

$$|\mathbb{E}\{F_S(\widehat{q}_{\mathrm{St}})\} - \mathbb{E}\{F_S(\widehat{q})\}| \le \overline{L}_F \mathbb{E}(|\widehat{q}_{\mathrm{St}} - \widehat{q}|). \tag{11}$$

Define $d_0$ as the $\infty$-Wasserstein distance and $d(\cdot, \cdot) = d_0^2(\cdot, \cdot)$. Let

$$\widehat{J}_\lambda(\theta) = d\left(\widehat{F}_S^0(\cdot), \widehat{F}_S^1(\cdot; F_\theta)\right) + \lambda \|\theta - \widehat{\theta}\|_2^2,$$

with

$$\widetilde{\theta} = \arg\min_{\theta \in \Theta} \widehat{J}_\lambda(\theta), \qquad \widetilde{\theta}_0 = \arg\min_{\theta \in \Theta} \widehat{J}_0(\theta).$$

Then $\widehat{J}_\lambda(\widetilde{\theta}) \le \widehat{J}_\lambda(\widehat{\theta}) \wedge \widehat{J}_\lambda(\widetilde{\theta}_0)$. By Assumption 4.1(ii), the support diameter of $\widehat{F}_S^0$ and $\widehat{F}_S^1(\cdot; F_\theta)$ is bounded by $M_\Theta < \infty$.

**Case 1:** $\widehat{J}_\lambda(\widetilde{\theta}) \le \widehat{J}_\lambda(\widehat{\theta})$ implies

$$\sqrt{\lambda}\|\widetilde{\theta} - \widehat{\theta}\|_2 \le d_0\left(\widehat{F}_S^0(\cdot), \widehat{F}_S^1(\cdot; F_{\widehat{\theta}})\right) \le M_\Theta.$$

Thus by (10),

$$|\mathbb{E}\{F_S(\widehat{q}_{\mathrm{St}})\} - \mathbb{E}\{F_S(\widehat{q}_0)\}| \le L_\Theta \overline{L}_F M_\Theta \lambda^{-1/2}.$$

By the definition of $\delta_S$ and the Lipschitz property of $F_S$,

$$|\mathbb{E}\{F_S(\widehat{q}_0)\} - (1 - \alpha)| \le \overline{L}_F \delta_S.$$

Therefore

$$\left|\mathbb{P}\left(Y_{n+1} \in \widehat{C}_{\mathrm{St}}(X_{n+1})\right) - (1 - \alpha)\right| \le \overline{L}_F \delta_S + L_\Theta \overline{L}_F M_\Theta \lambda^{-1/2} + n^{-1}. \tag{12}$$

**Case 2:** $\widehat{J}_\lambda(\widetilde{\theta}) \le \widehat{J}_\lambda(\widetilde{\theta}_0)$ implies

$$\begin{aligned}
d_0^2\left(\widehat{F}_S^0(\cdot), \widehat{F}_S^1(\cdot; F_{\widetilde{\theta}})\right) &\le d_0^2\left(\widehat{F}_S^0(\cdot), \widehat{F}_S^1(\cdot; F_{\widetilde{\theta}_0})\right) + \lambda\|\widetilde{\theta}_0 - \widehat{\theta}\|_2^2 \\
&\le \epsilon^2 + C_\Theta \lambda \\
&\le \left(\epsilon + C_\Theta^{1/2}\lambda^{1/2}\right)^2,
\end{aligned}$$

where $C_\Theta$ is the diameter bound of $\Theta$ and we use

$$\min_{\theta \in \Theta} d\left(\widehat{F}_S^0, \widehat{F}_S^1(\cdot; F_\theta)\right) \le \epsilon^2.$$

Since $d_0$ controls quantile differences,

$$|\widehat{q}_{\mathrm{St}} - \widehat{q}| \le d_0\left(\widehat{F}_S^0(\cdot), \widehat{F}_S^1(\cdot; F_{\widetilde{\theta}})\right) \le \epsilon + C_\Theta^{1/2}\lambda^{1/2}.$$

By (11),

$$|\mathbb{E}\{F_S(\widehat{q}_{\mathrm{St}})\} - \mathbb{E}\{F_S(\widehat{q})\}| \le \overline{L}_F \epsilon + \overline{L}_F C_\Theta^{1/2}\lambda^{1/2}.$$

Since $|\mathbb{E}\{F_S(\widehat{q})\} - (1 - \alpha)| \le n^{-1}$,

$$\left|\mathbb{P}\left(Y_{n+1} \in \widehat{C}_{\mathrm{St}}(X_{n+1})\right) - (1 - \alpha)\right| \le \overline{L}_F \epsilon + 2\overline{L}_F C_\Theta^{1/2}\lambda^{1/2} + n^{-1}. \tag{13}$$

Combining (12) and (13), there exists $C > 0$ such that

$$\left|\mathbb{P}\left(Y_{n+1} \in \widehat{C}_{\mathrm{St}}(X_{n+1})\right) - (1 - \alpha)\right| \le C \min(\epsilon + \lambda^{1/2} + n^{-1}, \delta_S + \lambda^{-1/2} + n^{-1}).$$

$\square$

## B.4. Proof of Lemma 4.4

*Proof.* By assumption, conditional on $\mathcal{D}_{\mathrm{aux}}$ there exists a deterministic map $L_0$ such that

$$L(\widehat{C}) = L_0(\widehat{q}), \qquad |L_0(q_1) - L_0(q_2)| \leq C_L|q_1 - q_2|.$$

Let $\widehat{q}'$ be an i.i.d. copy of $\widehat{q}$ conditional on $\mathcal{D}_{\mathrm{aux}}$. Then

$$\begin{aligned}
2\mathrm{Var}(L(\widehat{C}) \mid \mathcal{D}_{\mathrm{aux}}) &= \mathbb{E}\left[\{L_0(\widehat{q}) - L_0(\widehat{q}')\}^2 \mid \mathcal{D}_{\mathrm{aux}}\right] \\
&\leq C_L^2 \mathbb{E}\left[(\widehat{q} - \widehat{q}')^2 \mid \mathcal{D}_{\mathrm{aux}}\right] \\
&= 2C_L^2 \mathrm{Var}(\widehat{q} \mid \mathcal{D}_{\mathrm{aux}}),
\end{aligned}$$

which yields

$$\mathrm{Var}(L(\widehat{C}) \mid \mathcal{D}_{\mathrm{aux}}) = O(\mathrm{Var}(\widehat{q} \mid \mathcal{D}_{\mathrm{aux}})).$$

$\square$

## B.5. Proof of Theorem 4.6

*Proof.* Throughout this proof, all variances and expectations are conditional on $\mathcal{D}_{\mathrm{aux}}$, and we suppress this conditioning for brevity.

By Lemma 4.4, there exists $C_L > 0$ such that

$$\mathrm{Var}(L(\widehat{C})) \leq C_L^2 \mathrm{Var}(\widehat{q}), \qquad \mathrm{Var}(L(\widehat{C}_{\mathrm{St}})) \leq C_L^2 \mathrm{Var}(\widehat{q}_{\mathrm{St}}).$$

Therefore it is enough to bound $\mathrm{Var}(\widehat{q})$ and $\mathrm{Var}(\widehat{q}_{\mathrm{St}})$.

**Case 1: bound for** $\mathrm{Var}(\widehat{q})$. Since the density of $F_S$ is lower bounded by $\underline{L}_F$, the quantile map is $\underline{L}_F^{-1}$-Lipschitz:

$$|Q(u_1; F_S) - Q(u_2; F_S)| \leq \underline{L}_F^{-1}|u_1 - u_2|, \qquad u_1, u_2 \in (0, 1).$$

Let $\widehat{q} = Q(1 - \alpha_n; \widehat{F}_S^0)$ and define $U = F_S(\widehat{q})$. Under continuity of $F_S$ and the density lower bound in Assumption 4.1(iv), $U$ is an order statistic of $n$ i.i.d. uniforms, hence $U \sim \mathrm{Beta}(k, n + 1 - k)$ for some $k \in \{1, \ldots, n\}$ and $\mathrm{Var}(U) = O(n^{-1})$. Thus

$$\mathrm{Var}(\widehat{q}) = \mathrm{Var}(Q(U; F_S)) \leq \underline{L}_F^{-2}\mathrm{Var}(U) = O(n^{-1}),$$

and consequently $\mathrm{Var}(L(\widehat{C})) = O(n^{-1})$.

**Case 2: bound for** $\mathrm{Var}(\widehat{q}_{\mathrm{St}})$. Define

$$\widehat{Q}(\theta) = Q(1 - \alpha_n; \widehat{F}_S^1(\cdot; F_\theta)), \qquad \widetilde{Q}(\theta) = Q(1 - \alpha_n; F_S^1(\cdot; F_\theta)),$$

$$\begin{aligned}
\widehat{q}^0(u) &= Q(u; \widehat{F}_S^0), & q^0(u) &= Q(u; F_S), \\
\widehat{q}^1(u; \theta) &= Q(u; \widehat{F}_S^1(\cdot; F_\theta)), & q^1(u; \theta) &= Q(u; F_S^1(\cdot; F_\theta)).
\end{aligned}$$

where

$$F_S^1(s; F_\theta) = \mathbb{E}\{F(s \mid X; \theta)\},$$

whose expectation is taken over $X \sim P_X$. Let

$$\widehat{J}_\lambda(\theta) = d\left(\widehat{F}_S^0, \widehat{F}_S^1(\cdot; F_\theta)\right) + \lambda\|\theta - \widehat{\theta}\|_2^2, \qquad J_\lambda(\theta) = d\left(F_S, F_S^1(\cdot; F_\theta)\right) + \lambda\|\theta - \widehat{\theta}\|_2^2.$$

Therefore, define

$$\widehat{q}_{\mathrm{St}} = \widehat{Q}(\widetilde{\theta}), \qquad \widetilde{q}_{\mathrm{St}} = \widetilde{Q}(\widetilde{\theta}), \qquad \theta_\lambda = \arg\min_{\theta \in \Theta} J_\lambda(\theta).$$

Let $\widehat{q}'_{\mathrm{St}}$ and $\widetilde{q}'_{\mathrm{St}}$ be i.i.d. copies of $\widehat{q}_{\mathrm{St}}$ and $\widetilde{q}_{\mathrm{St}}$, respectively. Then

$$2\mathrm{Var}(\widehat{q}_{\mathrm{St}}) = \mathbb{E}(\widehat{q}_{\mathrm{St}} - \widehat{q}'_{\mathrm{St}})^2$$
$$\leq 3\mathbb{E}(\widetilde{q}_{\mathrm{St}} - \widetilde{q}'_{\mathrm{St}})^2 + 6\mathbb{E}(\widetilde{q}_{\mathrm{St}} - \widehat{q}_{\mathrm{St}})^2. \tag{14}$$

**Step 1 (control of $\mathbb{E}(\widetilde{q}_{\mathrm{St}} - \widetilde{q}'_{\mathrm{St}})^2$).** By the same quantile-Lipschitz argument used in Theorem 4.2 derived from Lemma B.1, there exists $L_\Theta > 0$ such that

$$|\widetilde{Q}(\theta_1) - \widetilde{Q}(\theta_2)| \leq L_\Theta \|\theta_1 - \theta_2\|_2.$$

Hence

$$\mathbb{E}(\widetilde{q}_{\mathrm{St}} - \widetilde{q}'_{\mathrm{St}})^2 = \mathbb{E}|\widetilde{Q}(\widetilde{\theta}) - \widetilde{Q}(\widetilde{\theta}')|^2 \leq L_\Theta^2 \mathbb{E}\|\widetilde{\theta} - \widetilde{\theta}'\|_2^2 = 2L_\Theta^2 \mathrm{tr}\{\mathrm{Cov}(\widetilde{\theta})\}.$$

It remains to bound $\mathrm{tr}\{\mathrm{Cov}(\widetilde{\theta})\}$. Define score maps

$$\widehat{\Psi}(\theta) = \nabla_\theta \widehat{J}_\lambda(\theta), \qquad \Psi(\theta) = \nabla_\theta J_\lambda(\theta).$$

Define

$$\widehat{H}_\lambda(\theta) = \nabla_{\theta\theta} \widehat{J}_\lambda(\theta), \qquad H_\lambda(\theta) = \nabla_{\theta\theta} J_\lambda(\theta).$$

A first-order Taylor expansion of $\widehat{\Psi}$ at $\theta_\lambda$ gives

$$0 = \widehat{\Psi}(\widetilde{\theta}) = \widehat{\Psi}(\theta_\lambda) + \widehat{H}_\lambda(\theta_\lambda)(\widetilde{\theta} - \theta_\lambda) + r_n,$$

where the remainder satisfies

$$\|r_n\|_2 \leq C\|\widetilde{\theta} - \theta_\lambda\|_2^2.$$

Therefore

$$\widetilde{\theta} - \theta_\lambda = -\widehat{H}_\lambda(\theta_\lambda)^{-1}\widehat{\Psi}(\theta_\lambda) + R_n,$$

with

$$R_n = -\widehat{H}_\lambda(\theta_\lambda)^{-1} r_n, \qquad \|R_n\|_2 \leq C\|\widetilde{\theta} - \theta_\lambda\|_2^2,$$

where we used the lower bound for $\widehat{H}_\lambda(\theta_\lambda)$ established below. Since $\Psi(\theta_\lambda) = 0$, we have $\widehat{\Psi}(\theta_\lambda) = \widehat{\Psi}(\theta_\lambda) - \Psi(\theta_\lambda)$.

By Assumption 4.5,

$$H_\lambda(\theta_\lambda) = \nabla_{\theta\theta} d\big(F_S, F_S^1(\cdot; F_{\theta_\lambda})\big) + 2\lambda I \succeq (c_d + 2\lambda)I.$$

If $c_d > 0$, then with $h_\lambda = 1 + \lambda$ and $c_H = \min\{c_d, 2\}$,

$$\lambda_{\min}\{H_\lambda(\theta_\lambda)\} \geq c_H h_\lambda.$$

If $c_d \leq 0$ and $\lambda \geq 1 - c_d$, then $c_d + 2\lambda \geq \lambda + 1 \geq \lambda$, so with $h_\lambda = \lambda$ and $c_H = 1$ we have

$$\lambda_{\min}\{H_\lambda(\theta_\lambda)\} \geq c_H h_\lambda.$$

To compare $\widehat{H}_\lambda(\theta_\lambda)$ and $H_\lambda(\theta_\lambda)$, note that for $d(\cdot, \cdot) = W_2^2(\cdot, \cdot)$,

$$\widehat{H}_\lambda(\theta_\lambda) = 2\int_0^1 \big[\nabla_\theta \widehat{q}^1(u; \theta_\lambda) \nabla_\theta \widehat{q}^1(u; \theta_\lambda)^\top + \{\widehat{q}^1(u; \theta_\lambda) - \widehat{q}^0(u)\}\nabla_{\theta\theta} \widehat{q}^1(u; \theta_\lambda)\big] du + 2\lambda I,$$

with the analogous population formula for $H_\lambda(\theta_\lambda)$. Hence each entry of $\widehat{H}_\lambda(\theta_\lambda) - H_\lambda(\theta_\lambda)$ is a finite sum of integrals involving pointwise differences of $\widehat{q}^0 - q^0$, $\widehat{q}^1 - q^1$, $\nabla_\theta \widehat{q}^1 - \nabla_\theta q^1$, and $\nabla_{\theta\theta}\widehat{q}^1 - \nabla_{\theta\theta} q^1$, all evaluated at the fixed point $\theta_\lambda$. The first three terms are controlled exactly as below. For the last term, differentiating the quantile identity once more shows that $\nabla_{\theta\theta}\widehat{q}^1(u; \theta_\lambda)$ is again a ratio of empirical averages of bounded quantities involving $\partial_\theta \widehat{F}_S^1$, $\partial_{\theta\theta}\widehat{F}_S^1$, $\partial_\theta \widehat{f}_S^1$, and $\partial_s \widehat{f}_S^1$. Under the same local smoothness used for the Taylor remainder, these empirical averages converge pointwise to their population counterparts, so for each fixed $u$ the resulting second-derivative difference is $o_p(1)$. Dominated convergence then gives entrywise $o_p(1)$ for the Hessian difference, and since the parameter dimension is fixed,

$$\|\widehat{H}_\lambda(\theta_\lambda) - H_\lambda(\theta_\lambda)\|_{\mathrm{op}} = o_p(1).$$

Therefore

$$\lambda_{\min}\{\widehat{H}_\lambda(\theta_\lambda)\} \geq c_H h_\lambda - o_p(1),$$

and hence

$$\|\widehat{H}_\lambda(\theta_\lambda)^{-1}\|_2 = O_p(h_\lambda^{-1}).$$

It remains to control the pointwise fluctuation $\widehat{\Psi}(\theta_\lambda) - \Psi(\theta_\lambda)$. For $d(\cdot, \cdot) = W_2^2(\cdot, \cdot)$, write

$$\widehat{\Psi}(\theta_\lambda) - \Psi(\theta_\lambda) = T_1 + T_2 + T_3,$$

where

$$T_1 = 2 \int_0^1 \{\widehat{q}^1(u; \theta_\lambda) - \widehat{q}^0(u)\}\{\nabla_\theta \widehat{q}^1(u; \theta_\lambda) - \nabla_\theta q^1(u; \theta_\lambda)\} \, du,$$

$$T_2 = 2 \int_0^1 \{\widehat{q}^1(u; \theta_\lambda) - q^1(u; \theta_\lambda)\}\nabla_\theta q^1(u; \theta_\lambda) \, du,$$

$$T_3 = -2 \int_0^1 \{\widehat{q}^0(u) - q^0(u)\}\nabla_\theta q^1(u; \theta_\lambda) \, du.$$

By the density lower bound in Assumption 4.1(iv), Lemma B.1, and the DKW inequality, for each $u \in (0, 1)$,

$$\mathbb{E}|\widehat{q}^0(u) - q^0(u)|^2 = O(n^{-1}), \qquad \mathbb{E}|\widehat{q}^1(u; \theta_\lambda) - q^1(u; \theta_\lambda)|^2 = O(m^{-1}).$$

Also,

$$\nabla_\theta q^1(u; \theta) = -\frac{\partial_\theta F_S^1(s; F_\theta)|_{s=q^1(u;\theta)}}{f_S^1(q^1(u; \theta); F_\theta)},$$

and the analogous identity holds for $\nabla_\theta \widehat{q}^1(u; \theta_\lambda)$. For fixed $u$, define

$$A_m(u) = \partial_\theta \widehat{F}_S^1(\widehat{q}^1(u; \theta_\lambda); F_{\theta_\lambda}), \qquad A(u) = \partial_\theta F_S^1(q^1(u; \theta_\lambda); F_{\theta_\lambda}),$$

$$B_m(u) = \widehat{f}_S^1(\widehat{q}^1(u; \theta_\lambda); F_{\theta_\lambda}), \qquad B(u) = f_S^1(q^1(u; \theta_\lambda); F_{\theta_\lambda}).$$

Since $\theta_\lambda$ is fixed after conditioning on $\mathcal{D}_{\text{aux}}$, the empirical averages defining $\widehat{F}_S^1$, $\partial_\theta \widehat{F}_S^1$, and $\widehat{f}_S^1$ are pointwise averages of bounded random variables. Therefore, for each $u$,

$$\mathbb{E}\|A_m(u) - A(u)\|_2^2 = O(m^{-1}), \qquad \mathbb{E}|B_m(u) - B(u)|^2 = O(m^{-1}).$$

Combining these bounds with the quantile rate and the ratio identity

$$\left\|\frac{A_m(u)}{B_m(u)} - \frac{A(u)}{B(u)}\right\|_2 \leq \underline{L}_F^{-1}\|A_m(u) - A(u)\|_2 + \underline{L}_F^{-2}\|A(u)\|_2 |B_m(u) - B(u)|$$

yields, for each $u$,

$$\mathbb{E}\|\nabla_\theta \widehat{q}^1(u; \theta_\lambda) - \nabla_\theta q^1(u; \theta_\lambda)\|_2^2 = O(m^{-1}).$$

Since $\nabla_\theta q^1(u; \theta_\lambda)$ and $\widehat{q}^1(u; \theta_\lambda) - \widehat{q}^0(u)$ are uniformly bounded in $u$ by Assumption 4.1(ii),(iv), integrating the preceding pointwise bounds over $u$ gives

$$\mathbb{E}\|T_3\|_2^2 = O(n^{-1}), \qquad \mathbb{E}\|T_2\|_2^2 = O(m^{-1}), \qquad \mathbb{E}\|T_1\|_2^2 = O(m^{-1}).$$

Hence

$$\mathbb{E}\|\widehat{\Psi}(\theta_\lambda) - \Psi(\theta_\lambda)\|_2^2 = O(n^{-1} + m^{-1}),$$

so the linear term in the expansion is of order $O_p(h_\lambda^{-1}(n^{-1/2} + m^{-1/2}))$. A standard fixed-point argument therefore gives

$$\widetilde{\theta} - \theta_\lambda = O_p\left(h_\lambda^{-1}(n^{-1/2} + m^{-1/2})\right), \qquad R_n = O_p\left(h_\lambda^{-2}(n^{-1} + m^{-1})\right).$$

Consequently, the second-moment order is

$$\mathbb{E}\|\widetilde{\theta} - \theta_\lambda\|_2^2 = O\big((n^{-1} + m^{-1})h_\lambda^{-2}\big),$$

and since

$$\mathrm{tr}\{\mathrm{Cov}(\widetilde{\theta})\} = \mathbb{E}\|\widetilde{\theta} - \mathbb{E}\widetilde{\theta}\|_2^2 \le \mathbb{E}\|\widetilde{\theta} - \theta_\lambda\|_2^2,$$

we have

$$\mathrm{tr}\{\mathrm{Cov}(\widetilde{\theta})\} = O\big((n^{-1} + m^{-1})h_\lambda^{-2}\big).$$

Therefore

$$\mathbb{E}(\widetilde{q}_{\mathrm{St}} - \widetilde{q}_{\mathrm{St}}')^2 = O\big((n^{-1} + m^{-1})h_\lambda^{-2}\big).$$

**Step 2 (control of $\mathbb{E}(\widehat{q}_{\mathrm{St}} - \widetilde{q}_{\mathrm{St}})^2$).** Decompose around $\theta_\lambda$:

$$\mathbb{E}(\widehat{q}_{\mathrm{St}} - \widetilde{q}_{\mathrm{St}})^2 \le 3\mathbb{E}\{\widehat{Q}(\widetilde{\theta}) - \widehat{Q}(\theta_\lambda)\}^2 + 3\mathbb{E}\{\widetilde{Q}(\widetilde{\theta}) - \widetilde{Q}(\theta_\lambda)\}^2$$
$$+ 3\mathbb{E}\{\widehat{Q}(\theta_\lambda) - \widetilde{Q}(\theta_\lambda)\}^2.$$

By the Lipschitz property of $\widehat{Q}$ and $\widetilde{Q}$ in $\theta$ and Step 1, the first two terms are

$$O\big((n^{-1} + m^{-1})h_\lambda^{-2}\big).$$

For the last term, fix $u_0 = 1 - \alpha_n$ and write

$$q_\lambda = q^1(u_0; \theta_\lambda), \qquad \widehat{q}_\lambda = \widehat{q}^1(u_0; \theta_\lambda).$$

Then

$$\widehat{Q}(\theta_\lambda) - \widetilde{Q}(\theta_\lambda) = \widehat{q}_\lambda - q_\lambda.$$

Since $\widehat{F}_S^1(\cdot; F_{\theta_\lambda})$ has density lower bounded by $\underline{L}_F$ (Assumption 4.1(iv)), monotonicity yields

$$|\widehat{q}_\lambda - q_\lambda| \le \underline{L}_F^{-1} \left|\widehat{F}_S^1(q_\lambda; F_{\theta_\lambda}) - F_S^1(q_\lambda; F_{\theta_\lambda})\right|.$$

Also,

$$\widehat{F}_S^1(q_\lambda; F_{\theta_\lambda}) = m^{-1} \sum_{j=1}^m F(q_\lambda \mid \widetilde{X}_j; \theta_\lambda),$$

whose summands lie in $[0, 1]$. Hence Hoeffding gives

$$\mathbb{P}\left(\left|\widehat{F}_S^1(q_\lambda; F_{\theta_\lambda}) - F_S^1(q_\lambda; F_{\theta_\lambda})\right| > t\right) \le 2e^{-2mt^2},$$

and therefore

$$\mathbb{E}\left|\widehat{F}_S^1(q_\lambda; F_{\theta_\lambda}) - F_S^1(q_\lambda; F_{\theta_\lambda})\right|^2 = O(m^{-1}).$$

Consequently,

$$\mathbb{E}\{\widehat{Q}(\theta_\lambda) - \widetilde{Q}(\theta_\lambda)\}^2 = \mathbb{E}(\widehat{q}_\lambda - q_\lambda)^2 = O(m^{-1}).$$

Thus

$$\mathbb{E}(\widehat{q}_{\mathrm{St}} - \widetilde{q}_{\mathrm{St}})^2 = O\big((n^{-1} + m^{-1})h_\lambda^{-2}\big) + O(m^{-1}).$$

Plugging these bounds into (14),

$$\mathrm{Var}(\widehat{q}_{\mathrm{St}}) = O\big((n^{-1} + m^{-1})h_\lambda^{-2}\big) + O(m^{-1}) = O\big(m^{-1} + n^{-1}h_\lambda^{-2}\big).$$

Hence

$$\mathrm{Var}(L(\widehat{C}_{\mathrm{St}})) = O\big(m^{-1} + n^{-1}h_\lambda^{-2}\big).$$

$\square$

*Remark* B.2. The proof of Theorem 4.6 also uses a local second-order expansion of $\widehat{\Psi}$ around $\theta_\lambda$. We do not state this as part of Assumption 4.5, since it is only a routine proof device: it follows, for example, if $\widehat{\Psi}$ is continuously differentiable in a neighborhood of $\theta_\lambda$ and its Jacobian is locally Lipschitz there. Under the same local smoothness that yields the pointwise Hessian consistency discussed in Appendix A.4, the remainder satisfies $\|r(\theta)\|_2 \le C\|\theta - \theta_\lambda\|_2^2$ locally.

## B.6. Proof of Theorem 4.7

*Proof.* Since $\widehat{q}_{\text{St-sel}} \in [q_{\text{L}}, q_{\text{U}}]$, monotonicity of threshold sets implies

$$\{y : S(X_{n+1}, y) \leq q_{\text{L}}\} \subseteq \widehat{C}_{\text{St-sel}}(X_{n+1}) \subseteq \{y : S(X_{n+1}, y) \leq q_{\text{U}}\}.$$

Therefore,

$$\mathbb{P}(S_{n+1} \leq q_{\text{L}}) \leq \mathbb{P}\Big(Y_{n+1} \in \widehat{C}_{\text{St-sel}}(X_{n+1})\Big) \leq \mathbb{P}(S_{n+1} \leq q_{\text{U}}).$$

For any $\beta \in (0, 1)$, let $q_\beta = Q(1 - \beta; \widehat{F}_S^0)$ and $k_\beta = \lceil n(1 - \beta) \rceil$. Then $q_\beta = S_{(k_\beta)}$, where $S_{(k)}$ is the $k$-th order statistic of $(S_1, \ldots, S_n)$. By exchangeability, the rank of $S_{n+1}$ among $(S_1, \ldots, S_n, S_{n+1})$ is uniform on $\{1, \ldots, n+1\}$, hence

$$\mathbb{P}(S_{n+1} \leq S_{(k_\beta)}) \in \left[\frac{k_\beta}{n+1}, \frac{k_\beta + 1}{n+1}\right).$$

Since $k_\beta = \lceil n(1 - \beta) \rceil$ implies

$$\frac{k_\beta}{n+1} \geq 1 - \beta, \qquad \frac{k_\beta + 1}{n+1} < 1 - \beta + \frac{1}{n+1},$$

we get

$$1 - \beta \leq \mathbb{P}(S_{n+1} \leq q_\beta) < 1 - \beta + \frac{1}{n+1}.$$

Using $\beta = \alpha + \alpha_{\text{tol}}$ gives

$$\mathbb{P}(S_{n+1} \leq q_{\text{L}}) \geq 1 - \alpha - \alpha_{\text{tol}},$$

and using $\beta = \alpha - \alpha_{\text{tol}}$ gives

$$\mathbb{P}(S_{n+1} \leq q_{\text{U}}) < 1 - \alpha + \alpha_{\text{tol}} + \frac{1}{n+1}.$$

Combining these inequalities proves the theorem. □

# C. Additional Experimental Results

This section provides supplementary experiments, elaborating on experimental details and presenting additional real-data details not fully covered in the main text.

## C.1. Implementation Details

For the "Std" metric, the calculation is performed across different repeats. Suppose a certain metric yields values $a_1, a_2, \ldots, a_{50}$ from 50 repeated runs. Then "Std" is computed as follows:

$$\text{Std} = \sqrt{\frac{1}{49} \sum_{i=1}^{50} (a_i - \bar{a})^2},$$

where $\bar{a}$ denotes the sample mean of the $a_1, a_2, \ldots, a_{50}$.

We next detail the specific calculation method for the percentage improvement reported for "ours" and "ours-sel" in the Std block of Table 1. Denote by $a_1$ the Std value of the reported StCP variant (either "ours" or "ours-sel") and $a_0$ the Std value of the corresponding oracle baseline. Let $a_{\text{ref}}$ denote the reference baseline, defined as the smallest Std among "base", SDCP, and PPI whose marginal coverage remains within the acceptable range indicated in the Marginal block of Table 1. The reported improvement is calculated as

$$\left(1 - \frac{a_1 - a_0}{a_{\text{ref}} - a_0}\right) \times 100\%.$$

For clarity, we next detail the computation procedures for the corresponding base and oracle baselines. For the base method, the source-trained estimator used by the underlying GLCP-type or CQR-type procedure is obtained in the same way as

in StCP using $\mathcal{A}(\mathcal{L}'_N)$, after which the prediction set is constructed directly from $\mathcal{L}_n$ without the transductive alignment step. For the oracle baseline, we apply the same corresponding base procedure after augmenting calibration with additional labeled target data unavailable to StCP. This construction reduces the randomness caused by the limited target calibration set and thus serves as a meaningful upper benchmark.

For the direct plug-in baseline, the estimator $\widehat{F}_{S|X}$ is again obtained by $\mathcal{A}(\mathcal{L}'_N)$. The marginal CDF is then approximated by $\widehat{F}^1_S$ without alignment or debiasing, and the final prediction set is constructed as $\widehat{C}_{\mathrm{DP}} = \{y : S(X_{n+1}, y) \leq \widehat{q}_{\mathrm{DP}}\}$.

For StCP, we compute results for all $\lambda$ values within a predefined range selected via a grid search and report the set stability and other metrics corresponding to the $\lambda$ that yields the best Std result while maintaining marginal coverage. In the robustness analysis of $\lambda$, we also report median performance metrics across all $\lambda$ values rather than only the optimal one.

In solving the optimization problem in (7), we directly apply gradient descent for the tuned parameter.

**Finite-grid Wasserstein alignment in implementation.** The theoretical formulation with Wasserstein distance aligns the full quantile functions of $\widehat{F}^0_S$ and $\widehat{F}^1_S(\cdot; \widetilde{F}_{S|X})$. In implementation, we approximate this objective on a finite grid of $K$ quantile levels and explicitly include the target level $1 - \alpha_n$ in that grid. Thus, at $\lambda = 0$, the optimization aligns these $K$ grid points, in particular the point corresponding to the target conformal quantile. In our experiments this finite-grid alignment is practically attainable because the number of tuned parameters is large relative to $K$: we fine-tune a $100 \times 100$ weight matrix, giving $k_0 = 10000 \gg K$. This is why $\lambda = 0$ is feasible in the implemented selection rule.

## C.2. Simulation Experiment Details and Additional Results

We consider three representative synthetic regression settings whose noise levels are determined by three different sigma families: Quad, Softplus, and LogAbs. In all settings, the nominal coverage level is fixed at $1 - \alpha = 0.9$, the target noise scale is fixed at $\gamma_t = 1$, and the source noise scale is set to $\gamma_s = 1.2$. We use the same predictor and conditional-distribution estimators as in the real-data experiments, with hidden layers $(50, 100, 100, 50)$, $d = 5$, and 50 repeated runs.

The synthetic source and target covariates are generated from shifted Gaussian distributions. Writing $\mathbf{1}_d$ for the $d$-dimensional all-one vector, we set $\mu_s = 0$ and $\mu_t = \mathbf{1}_d/2\sqrt{d}$, and generate

$$X' \sim N(\mu_s, I_d), \qquad Y' = \frac{3}{d} \sum_{j=1}^{d} X'_j + \varepsilon', \qquad \varepsilon' \mid X' \sim N\left(0, \sigma^2(X'; \gamma_s, \eta)\right),$$

$$X \sim N(\mu_t, I_d), \qquad Y = \frac{2}{d} \sum_{j=1}^{d} X_j + \varepsilon, \qquad \varepsilon \mid X \sim N\left(0, \sigma^2(X; \gamma_t, \eta)\right),$$

where $\eta$ denotes the sigma family. For the three displayed settings, the heteroscedastic scale functions are

$$\sigma_{\mathrm{quad}}(x; \gamma) = \sqrt{\gamma} \frac{\sum_{j=1}^{d} x_j^2}{\sqrt{d}},$$

$$\sigma_{\mathrm{softplus}}(x; \gamma) = \sqrt{\gamma} \frac{\sum_{j=1}^{d} \log(1 + e^{x_j})}{\sqrt{d}},$$

$$\sigma_{\mathrm{logabs}}(x; \gamma) = \sqrt{\gamma} \frac{\sum_{j=1}^{d} \log(1 + |x_j|)}{\sqrt{d}}.$$

In each repetition, all source and target samples are generated independently from the corresponding laws above; the oracle benchmark uses an additional labeled target sample of size 2000.

For the robustness study in $\lambda$, we fix $(n, m) = (30, 500)$ and evaluate the full grid of candidate values used by StCP. For the sample-size studies, we vary $n$ in $\{30, 100, 500\}$ with $m = 500$, and vary $m$ in $\{30, 100, 500\}$ with $n = 30$. In the tables below, "ours" reports the best Std result over the $\lambda$ grid while maintaining marginal coverage, while "ours-sel" reports the result of the data-driven parameter-selection rule from Section 3.1. In the Std block, the percentages shown for these two columns are always computed relative to the corresponding base method. Boldface in the Std and Size blocks is assigned by comparing against the baseline methods whose marginal coverage remains within the displayed acceptable range.

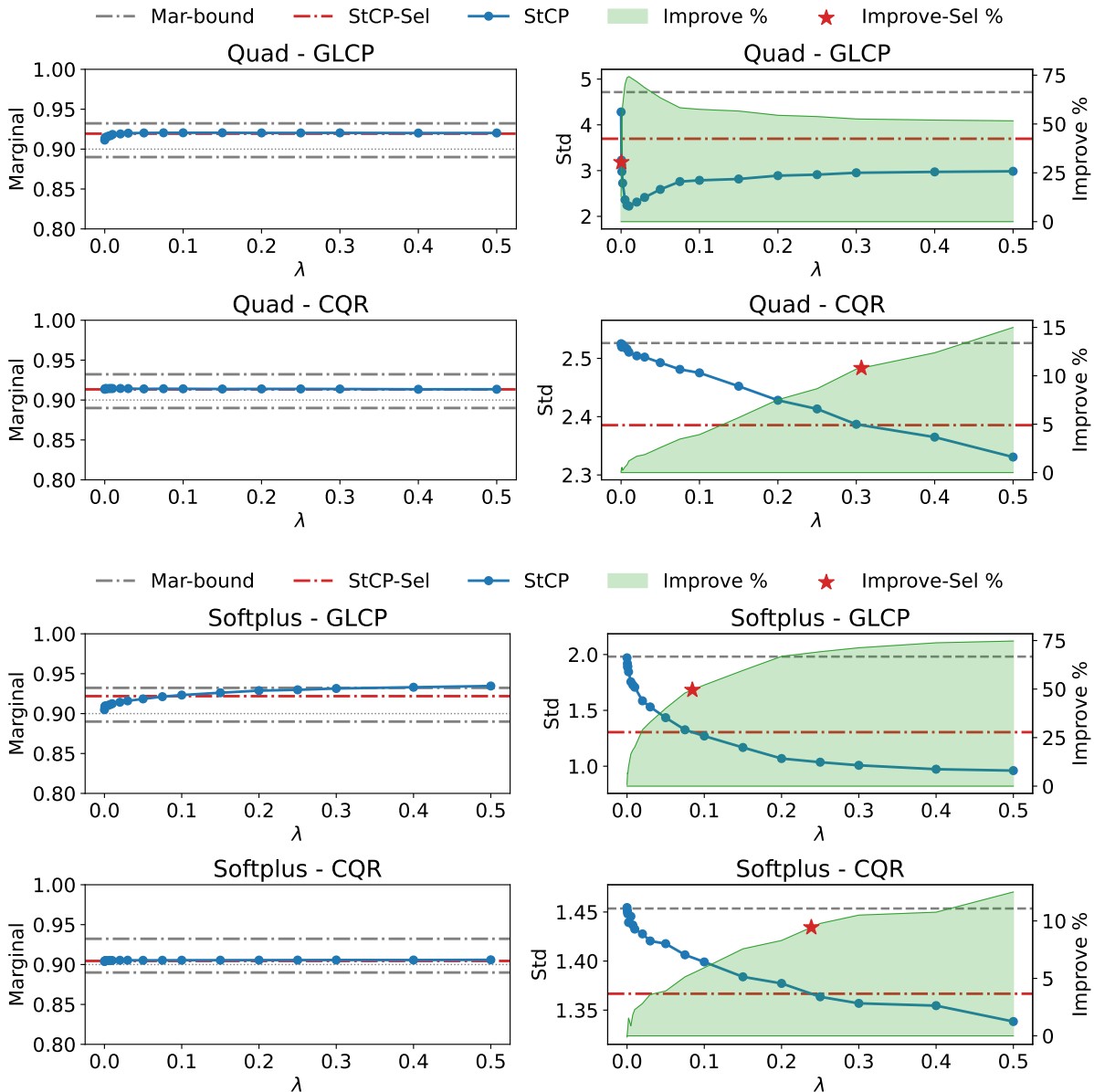

*Figure 4.* Additional $\lambda$ sensitivity results under the Quad and Softplus settings. The Quad setting exhibits the strongest variance reduction for the GLCP-type method, while the Softplus setting follows a smoother trajectory.

Figure 4 complements the main-text LogAbs plot. The Quad setting delivers the clearest improvement for the GLCP-type method, whereas the Softplus setting provides a smoother and more conservative trajectory. In both settings, the CQR-type method tends to benefit less than the GLCP-type method, but the direction of improvement remains broadly aligned whenever the marginal coverage stays acceptable.

*Table 3.* Simulation results with $m = 500$ and $n \in \{30, 100, 500\}$ under the Quad and Softplus settings. In the Marginal block, "$-$" indicates marginal coverage below $1 - \alpha - 0.01$, and "$+$" indicates marginal coverage above $1 - \alpha + 1/(n+1)$.

| | Setting | $n$ | Model | base | SDCP | PPI | ours | ours-sel | oracle | DP |
|---|---|---|---|---|---|---|---|---|---|---|
| | | | | | | | | Method | | |
| Std | Quad | 30 | GLCP | 4.72 | 6.71 | 4.51 | **2.22** (52.8%) | **3.70** (21.6%) | 1.36 | 3.06 |
| | | | CQR | 2.53 | 2.63 | 2.44 | **2.33** (7.7%) | **2.39** (5.6%) | 1.22 | 3.08 |
| | Quad | 100 | GLCP | 1.86 | $1.83^-$ | 1.97 | **0.79** (57.6%) | **1.38** (25.8%) | 0.55 | $0.82^+$ |
| | | | CQR | 1.24 | $1.13^-$ | 1.21 | **1.05** (14.9%) | **1.05** (14.9%) | 0.44 | $0.77^+$ |
| | Quad | 500 | GLCP | 0.69 | $0.55^-$ | $0.82^+$ | **0.41** (40.6%) | $0.41^+$ (41.4%) | $0.45^+$ | $0.45^+$ |
| | | | CQR | 0.46 | 0.52 | $0.44^+$ | **0.41** (9.2%) | **0.41** (9.2%) | 0.32 | $0.42^+$ |
| | Softplus | 30 | GLCP | 1.98 | 2.66 | 1.94 | **1.01** (49.1%) | **1.30** (34.1%) | 0.62 | $1.26^+$ |
| | | | CQR | 1.45 | 1.55 | 1.38 | **1.34** (7.9%) | **1.37** (6.0%) | 0.53 | $1.25^+$ |
| | Softplus | 100 | GLCP | 0.81 | $1.09^-$ | $0.80^+$ | **0.64** (20.9%) | $0.62^+$ (23.2%) | 0.24 | $0.70^+$ |
| | | | CQR | 0.69 | $0.84^-$ | 0.68 | **0.62** (10.5%) | **0.62** (10.7%) | 0.21 | $0.68^+$ |
| | Softplus | 500 | GLCP | 0.36 | $0.36^-$ | 0.39 | 0.38 (−2.9%) | $0.37^+$ (−0.7%) | $0.21^+$ | $0.33^+$ |
| | | | CQR | 0.31 | $0.31^-$ | 0.32 | **0.29** (8.4%) | **0.29** (8.1%) | 0.19 | $0.29^+$ |
| Marginal | Quad | 30 | GLCP | 0.914 | 0.916 | 0.917 | 0.918 | 0.919 | 0.902 | 0.921 |
| | | | CQR | 0.913 | 0.917 | 0.918 | 0.914 | 0.913 | 0.900 | 0.922 |
| | Quad | 100 | GLCP | 0.902 | $0.858^-$ | 0.905 | 0.909 | 0.906 | 0.902 | $0.915^+$ |
| | | | CQR | 0.897 | $0.888^-$ | 0.900 | 0.895 | 0.895 | 0.901 | $0.913^+$ |
| | Quad | 500 | GLCP | 0.901 | $0.873^-$ | $0.903^+$ | 0.901 | $0.907^+$ | $0.903^+$ | $0.910^+$ |
| | | | CQR | 0.902 | 0.891 | $0.902^+$ | 0.898 | 0.898 | 0.902 | $0.909^+$ |
| | Softplus | 30 | GLCP | 0.908 | 0.926 | 0.915 | 0.931 | 0.922 | 0.901 | $0.943^+$ |
| | | | CQR | 0.904 | 0.907 | 0.909 | 0.906 | 0.904 | 0.898 | $0.938^+$ |
| | Softplus | 100 | GLCP | 0.906 | $0.865^-$ | $0.910^+$ | 0.909 | $0.918^+$ | 0.902 | $0.935^+$ |
| | | | CQR | 0.898 | $0.888^-$ | 0.900 | 0.899 | 0.899 | 0.900 | $0.931^+$ |
| | Softplus | 500 | GLCP | 0.898 | $0.869^-$ | 0.902 | 0.901 | $0.916^+$ | $0.903^+$ | $0.934^+$ |
| | | | CQR | 0.900 | $0.886^-$ | 0.900 | 0.900 | 0.900 | 0.900 | $0.931^+$ |
| Size | Quad | 30 | GLCP | **12.81** | 14.56 | 13.13 | **12.21** | **12.80** | 10.70 | 12.92 |
| | | | CQR | **12.02** | 12.16 | 12.15 | **11.90** | **11.92** | 10.90 | 12.46 |
| | Quad | 100 | GLCP | **9.92** | $8.28^-$ | 10.27 | 10.05 | 9.99 | 9.57 | $10.56^+$ |
| | | | CQR | **9.92** | $9.60^-$ | 10.03 | **9.81** | **9.81** | 9.92 | $10.41^+$ |
| | Quad | 500 | GLCP | **9.20** | $8.04^-$ | $9.42^+$ | **9.14** | $9.44^+$ | $9.23^+$ | $9.80^+$ |
| | | | CQR | 9.58 | **9.22** | $9.59^+$ | 9.42 | 9.42 | 9.55 | $9.81^+$ |
| | Softplus | 30 | GLCP | **9.11** | 10.08 | 9.36 | 9.61 | 9.33 | 8.46 | $10.36^+$ |
| | | | CQR | **8.96** | 9.09 | 9.07 | 8.97 | **8.95** | 8.56 | $10.04^+$ |
| | Softplus | 100 | GLCP | **8.18** | $7.31^-$ | $8.35^+$ | 8.24 | $8.52^+$ | 7.97 | $9.19^+$ |
| | | | CQR | **8.15** | $7.97^-$ | 8.20 | 8.16 | 8.16 | 8.12 | $9.03^+$ |
| | Softplus | 500 | GLCP | **7.71** | $7.07^-$ | 7.84 | 7.79 | $8.22^+$ | $7.84^+$ | $8.87^+$ |
| | | | CQR | **7.97** | $7.67^-$ | 7.98 | **7.96** | **7.97** | 7.95 | $8.78^+$ |
| Miscoverage | Quad | 30 | GLCP | 0.019 | 0.023 | 0.023 | 0.023 | 0.023 | 0.017 | 0.026 |
| | | | CQR | 0.025 | 0.025 | 0.026 | 0.024 | 0.024 | 0.024 | 0.027 |
| | Quad | 100 | GLCP | 0.017 | 0.044 | 0.020 | 0.019 | 0.018 | 0.018 | 0.023 |
| | | | CQR | 0.024 | 0.027 | 0.023 | 0.024 | 0.024 | 0.023 | 0.023 |
| | Quad | 500 | GLCP | 0.018 | 0.031 | 0.020 | 0.018 | 0.018 | 0.018 | 0.022 |

Table 3 – continued from previous page

| | | | Method | | | | | | |
|---|---|---|---|---|---|---|---|---|---|
| Setting | $n$ | Model | base | SDCP | PPI | ours | ours-sel | oracle | DP |
| | | CQR | 0.023 | 0.025 | 0.023 | 0.023 | 0.023 | 0.023 | 0.022 |
| Softplus | 30 | GLCP | 0.016 | 0.028 | 0.021 | 0.032 | 0.024 | 0.016 | 0.043 |
| | | CQR | 0.022 | 0.022 | 0.022 | 0.022 | 0.022 | 0.023 | 0.039 |
| Softplus | 100 | GLCP | 0.015 | 0.036 | 0.019 | 0.016 | 0.021 | 0.015 | 0.036 |
| | | CQR | 0.023 | 0.026 | 0.023 | 0.023 | 0.023 | 0.023 | 0.033 |
| Softplus | 500 | GLCP | 0.015 | 0.033 | 0.018 | 0.015 | 0.020 | 0.015 | 0.035 |
| | | CQR | 0.023 | 0.027 | 0.023 | 0.023 | 0.023 | 0.023 | 0.033 |

*Table 4.* Simulation results with $n = 30$ and $m \in \{30, 100, 500\}$ across three representative sigma settings. In the Marginal block, "$-$" indicates marginal coverage below $1 - \alpha - 0.01$, and "$+$" indicates marginal coverage above $1 - \alpha + 1/(n + 1)$.

| | | | | Method | | | | | | |
|---|---|---|---|---|---|---|---|---|---|---|
| | Setting | $m$ | Model | base | SDCP | PPI | ours | ours-sel | oracle | DP |
| Std | Quad | 30 | GLCP | 4.72 | 5.93 | 4.67 | **2.43** (48.4%) | **3.83** (18.7%) | 1.25 | 3.46 |
| | | | CQR | 2.53 | 2.70 | 2.42 | 2.47 (2.1%) | 2.50 (1.0%) | 1.19 | 3.20 |
| | Quad | 100 | GLCP | 4.72 | 6.69 | 4.52 | **2.14** (54.6%) | **3.69** (21.7%) | 1.25 | 2.90 |
| | | | CQR | 2.53 | 2.69 | 2.41 | **2.40** (5.0%) | 2.44 (3.2%) | 1.22 | 2.76 |
| | Quad | 500 | GLCP | 4.72 | 6.71 | 4.51 | **2.22** (52.8%) | **3.70** (21.6%) | 1.36 | 3.06 |
| | | | CQR | 2.53 | 2.63 | 2.44 | **2.33** (7.7%) | **2.39** (5.6%) | 1.22 | 3.08 |
| | Softplus | 30 | GLCP | 1.98 | 2.18 | 1.99 | **1.35** (31.8%) | **1.50** (24.5%) | 0.64 | $1.53^{+}$ |
| | | | CQR | 1.45 | 1.42 | 1.38 | 1.40 (3.8%) | 1.42 (2.5%) | 0.54 | $1.34^{+}$ |
| | Softplus | 100 | GLCP | 1.98 | 2.29 | 1.96 | **1.02** (48.6%) | **1.32** (33.2%) | 0.60 | $1.30^{+}$ |
| | | | CQR | 1.45 | 1.44 | 1.44 | **1.38** (5.2%) | **1.40** (4.0%) | 0.51 | $1.24^{+}$ |
| | Softplus | 500 | GLCP | 1.98 | 2.66 | 1.94 | **1.01** (49.1%) | **1.30** (34.1%) | 0.62 | $1.26^{+}$ |
| | | | CQR | 1.45 | 1.55 | 1.38 | **1.34** (7.9%) | **1.37** (6.0%) | 0.53 | $1.25^{+}$ |
| | LogAbs | 30 | GLCP | 1.12 | 1.61 | 1.12 | **0.91** (18.2%) | **1.03** (7.7%) | 0.34 | 1.22 |
| | | | CQR | 0.98 | 0.93 | 0.95 | **0.92** (5.6%) | **0.93** (5.3%) | 0.30 | 1.05 |
| | LogAbs | 100 | GLCP | 1.12 | 1.64 | 1.11 | **0.77** (31.4%) | **0.89** (20.2%) | 0.31 | 0.86 |
| | | | CQR | 0.98 | 0.94 | 0.94 | **0.87** (11.1%) | **0.87** (11.0%) | 0.28 | 0.82 |
| | LogAbs | 500 | GLCP | 1.12 | $1.65^{+}$ | 1.09 | **0.77** (31.2%) | **0.88** (20.7%) | 0.36 | 0.85 |
| | | | CQR | 0.98 | 0.88 | 0.94 | **0.82** (16.3%) | **0.83** (15.4%) | 0.31 | 0.81 |
| Marginal | Quad | 30 | GLCP | 0.914 | 0.924 | 0.915 | 0.918 | 0.920 | 0.902 | 0.922 |
| | | | CQR | 0.913 | 0.913 | 0.917 | 0.915 | 0.915 | 0.901 | 0.926 |
| | Quad | 100 | GLCP | 0.914 | 0.927 | 0.916 | 0.913 | 0.918 | 0.901 | 0.912 |
| | | | CQR | 0.913 | 0.916 | 0.918 | 0.915 | 0.915 | 0.900 | 0.912 |
| | Quad | 500 | GLCP | 0.914 | 0.916 | 0.917 | 0.918 | 0.919 | 0.902 | 0.921 |
| | | | CQR | 0.913 | 0.917 | 0.918 | 0.914 | 0.913 | 0.900 | 0.922 |
| | Softplus | 30 | GLCP | 0.908 | 0.929 | 0.912 | 0.931 | 0.922 | 0.902 | $0.952^{+}$ |
| | | | CQR | 0.904 | 0.910 | 0.908 | 0.908 | 0.907 | 0.899 | $0.946^{+}$ |
| | Softplus | 100 | GLCP | 0.908 | 0.929 | 0.913 | 0.932 | 0.922 | 0.902 | $0.943^{+}$ |
| | | | CQR | 0.904 | 0.909 | 0.907 | 0.907 | 0.906 | 0.899 | $0.935^{+}$ |
| | Softplus | 500 | GLCP | 0.908 | 0.926 | 0.915 | 0.931 | 0.922 | 0.901 | $0.943^{+}$ |
| | | | CQR | 0.904 | 0.907 | 0.909 | 0.906 | 0.904 | 0.898 | $0.938^{+}$ |
| | LogAbs | 30 | GLCP | 0.899 | 0.927 | 0.904 | 0.910 | 0.910 | 0.905 | 0.928 |
| | | | CQR | 0.899 | 0.891 | 0.903 | 0.906 | 0.905 | 0.900 | 0.926 |
| | LogAbs | 100 | GLCP | 0.899 | 0.929 | 0.906 | 0.905 | 0.907 | 0.906 | 0.915 |

Table 4 – continued from previous page

| Setting | | $m$ | Model | base | SDCP | PPI | ours | ours-sel | oracle | DP |
|---|---|---|---|---|---|---|---|---|---|---|
| | | | | | | | | | | Method |
| | LogAbs | 500 | CQR | 0.899 | 0.897 | 0.905 | 0.904 | 0.904 | 0.901 | 0.907 |
| | | | GLCP | 0.899 | $0.935^+$ | 0.909 | 0.911 | 0.910 | 0.904 | 0.921 |
| | | | CQR | 0.899 | 0.896 | 0.905 | 0.903 | 0.903 | 0.899 | 0.916 |
| Size | Quad | 30 | GLCP | **12.81** | 14.31 | 13.06 | **12.26** | 12.87 | 10.63 | 13.32 |
| | | | CQR | **12.02** | 12.07 | 12.11 | 12.08 | 12.08 | 10.92 | 12.79 |
| | Quad | 100 | GLCP | **12.81** | 15.60 | 13.06 | **11.77** | **12.71** | 10.62 | **12.29** |
| | | | CQR | **12.02** | 12.18 | 12.17 | **12.00** | **12.01** | 10.91 | **11.94** |
| | Quad | 500 | GLCP | **12.81** | 14.56 | 13.13 | **12.21** | **12.80** | 10.70 | 12.92 |
| | | | CQR | **12.02** | 12.16 | 12.15 | **11.90** | **11.92** | 10.90 | 12.46 |
| | Softplus | 30 | GLCP | **9.11** | 9.95 | 9.31 | 9.67 | 9.40 | 8.50 | $10.97^+$ |
| | | | CQR | **8.96** | 9.14 | 9.06 | 9.04 | 9.03 | 8.59 | $10.43^+$ |
| | Softplus | 100 | GLCP | **9.11** | 10.02 | 9.30 | 9.62 | 9.34 | 8.49 | $10.37^+$ |
| | | | CQR | **8.96** | 9.11 | 9.04 | 9.02 | 9.00 | 8.59 | $9.92^+$ |
| | Softplus | 500 | GLCP | **9.11** | 10.08 | 9.36 | 9.61 | 9.33 | 8.46 | $10.36^+$ |
| | | | CQR | **8.96** | 9.09 | 9.07 | 8.97 | **8.95** | 8.56 | $10.04^+$ |
| | LogAbs | 30 | GLCP | **5.19** | 5.95 | 5.32 | 5.39 | 5.41 | 5.04 | 5.99 |
| | | | CQR | 5.24 | **5.09** | 5.30 | 5.32 | 5.31 | 5.04 | 5.80 |
| | LogAbs | 100 | GLCP | **5.19** | 6.01 | 5.35 | 5.24 | 5.29 | 5.07 | 5.54 |
| | | | CQR | 5.24 | **5.18** | 5.32 | 5.28 | 5.28 | 5.06 | 5.33 |
| | LogAbs | 500 | GLCP | **5.19** | $6.20^+$ | 5.39 | 5.36 | 5.34 | 5.04 | 5.65 |
| | | | CQR | 5.24 | **5.14** | 5.32 | 5.24 | 5.24 | 5.03 | 5.48 |
| Miscoverage | Quad | 30 | GLCP | 0.019 | 0.027 | 0.022 | 0.022 | 0.023 | 0.017 | 0.027 |
| | | | CQR | 0.025 | 0.024 | 0.025 | 0.025 | 0.025 | 0.024 | 0.030 |
| | Quad | 100 | GLCP | 0.019 | 0.029 | 0.023 | 0.020 | 0.022 | 0.017 | 0.022 |
| | | | CQR | 0.025 | 0.025 | 0.026 | 0.025 | 0.025 | 0.024 | 0.023 |
| | Quad | 500 | GLCP | 0.019 | 0.023 | 0.023 | 0.023 | 0.023 | 0.017 | 0.026 |
| | | | CQR | 0.025 | 0.025 | 0.026 | 0.024 | 0.024 | 0.024 | 0.027 |
| | Softplus | 30 | GLCP | 0.016 | 0.030 | 0.020 | 0.031 | 0.024 | 0.016 | 0.052 |
| | | | CQR | 0.022 | 0.023 | 0.022 | 0.022 | 0.022 | 0.023 | 0.047 |
| | Softplus | 100 | GLCP | 0.016 | 0.030 | 0.020 | 0.032 | 0.024 | 0.016 | 0.043 |
| | | | CQR | 0.022 | 0.022 | 0.022 | 0.022 | 0.022 | 0.023 | 0.036 |
| | Softplus | 500 | GLCP | 0.016 | 0.028 | 0.021 | 0.032 | 0.024 | 0.016 | 0.043 |
| | | | CQR | 0.022 | 0.022 | 0.022 | 0.022 | 0.022 | 0.023 | 0.039 |
| | LogAbs | 30 | GLCP | 0.014 | 0.028 | 0.016 | 0.016 | 0.016 | 0.015 | 0.029 |
| | | | CQR | 0.019 | 0.022 | 0.019 | 0.019 | 0.019 | 0.020 | 0.028 |
| | LogAbs | 100 | GLCP | 0.014 | 0.030 | 0.016 | 0.015 | 0.015 | 0.015 | 0.020 |
| | | | CQR | 0.019 | 0.020 | 0.019 | 0.019 | 0.019 | 0.020 | 0.019 |
| | LogAbs | 500 | GLCP | 0.014 | 0.035 | 0.017 | 0.017 | 0.016 | 0.015 | 0.023 |
| | | | CQR | 0.019 | 0.020 | 0.019 | 0.019 | 0.019 | 0.020 | 0.021 |

Tables 3 and 4 further confirm the main qualitative pattern. When $m$ is fixed and $n$ increases, the stability gain of StCP becomes smaller, which is consistent with the fact that auxiliary information is most useful when the labeled calibration sample is scarce. When $n$ is fixed and $m$ increases, the improvement generally strengthens from $m = 30$ to $m = 100$ and then stabilizes by $m = 500$, indicating diminishing returns from additional unlabeled target data. This pattern is especially visible under Quad and Softplus.

## C.3. Real Data Experiment Details

We present the data partitioning strategy and supplementary experimental details omitted from the main text. Among the MedMNIST benchmarks, our experiments include DermaMNIST (DERMA) and TissueMNIST (TISSUE), where TISSUE is the largest 2D subset with over 200K samples, and serves as a suitable large-scale classification dataset for our setting. As shown in Yang et al. (2021), standard baselines achieve relatively low accuracy on these two datasets, making them a meaningful testbed for evaluating conformal prediction stability.

**Data partitioning strategy.** For the BIO dataset (feature dimension $d = 9$, response: size of the residue), the features themselves do not exhibit clear distinctions for agent partitioning. However, domain-specific scenarios may lead to systematic variations—for instance, certain protein studies may involve residues with atypically large sizes. To capture such distributional differences, we adopt a response-based partitioning scheme. Given a dataset $\{(x_1, y_1), \ldots, (x_n, y_n)\}$, we first compute the $\alpha_0$-quantile $y_0$ of the response variable. Each sample $(x_i, y_i)$ is then assigned a weight proportional to $K(|y_i - y_0|/\sigma_0)$, where $K(\cdot)$ is the Gaussian kernel function and $\sigma_0$ scales the deviation. Finally, we perform probability-proportional sampling without replacement to construct the agent-specific dataset. To induce meaningful distribution shifts across agents, we set the quantile parameter $\alpha_0$ to 0.9 for BIO. The scale parameter $\sigma_0$ is defined as the difference between the $(\alpha_0 + 0.1)$ and $(\alpha_0 - 0.1)$ quantiles of the response variable, ensuring adaptive bandwidth for kernel weighting.

For the CRIME dataset ($d = 15$, response: violent crimes per 100K population per district), where existing research primarily focuses on densely populated metropolitan areas and leaves less-populated regions understudied, we designate the district with the smallest population as the target agent and randomly partition the others. In the STAR dataset ($d = 15$, response: ACT score), given that many studies examine the impact of early education on future achievement—particularly the negative effects of resource deprivation—we define students from rural schools in their early age as the target agent and randomly partition the remaining students.

For the classification dataset DERMA, the input image format for each sample is $28 \times 28 \times 3$. There are a total of 7 classes, and the class distribution is highly imbalanced, with some classes accounting for a very small proportion. To simulate disease-susceptibility prediction for a particular high-risk group, where such a group contains a relatively higher proportion of rare classes, we sample from all classes as uniformly as possible to form the target data pool, while the remaining samples constitute the auxiliary data pool. When partitioning the auxiliary data, it is divided evenly according to the number of agents.

For the classification dataset TISSUE, the input image format for each sample is $28 \times 28 \times 1$, and there are a total of 8 classes. Similar to DERMA, we construct the target data pool by sampling from all classes as uniformly as possible, while the remaining samples form the auxiliary data pool. The auxiliary data are then partitioned evenly according to the number of agents.

**Conditional coverage calculation.** Computing conditional coverage for prediction sets requires knowing the exact conditional distribution of $Y$ given $X$, which is unavailable in real-world experiments. For each test sample $(X, Y)$ and prediction set $\widehat{C}(X)$, we evaluate coverage via the indicator $\mathbb{I}(Y \in \widehat{C}(X))$. We evaluate our method using relaxed conditional coverage. For regression datasets, we partition the feature space into $N_x$ disjoint subspaces, ensuring each test sample belongs to exactly one subspace. Over repeated trials, we compute the average coverage rate within each subspace as the conditional coverage metric. Let $p_1, \ldots, p_{N_x}$ denote the conditional coverage rates and $w_1, \ldots, w_{N_x}$ the proportions of samples in each subspace. The conditional miscoverage is then defined as $\sum_{i=1}^{N_x} w_i |p_i - (1 - \alpha)|$. The partitioning strategy in this work first applies K-means clustering to the target agent's data to identify $N_x$ centroids, then divides the spatial domain into $N_x$ disjoint sub-regions based on these centroids, ensuring spatially coherent and interpretable partitions while maintaining alignment with the underlying data distribution. In our experiments we take $N_x = 10$.

For classification datasets, the test-conditional coverage is directly partitioned based on the class labels of the samples. Assume there exist $N_y$ classes for the dataset, and let $p_1, \ldots, p_{N_y}$ denote the conditional coverage rates and $w_1, \ldots, w_{N_y}$ the proportions of samples in each class. The gap between the coverage and $1 - \alpha$ is computed separately for each class, and then weighted by the proportion of each class to obtain the conditional miscoverage $\sum_{i=1}^{N_y} w_i |p_i - (1 - \alpha)|$.

