# OpenReview forum: "Stable Localized Conformal Prediction via Transduction"
_ICML.cc/2026/Conference — ICML 2026 regular_

### Official Review · Reviewer_Pd6D · 2026-02-26

**Soundness:** 3
**Presentation:** 2
**Significance:** 3
**Originality:** 3
**Overall Recommendation:** 3
**Confidence:** 3

**Summary:**

This paper introduces set stability to measure how sensitive conformal prediction set sizes are to the particular calibration sample, a problem that is especially severe for localized conformal methods with small calibration sets. It proposes Stable Localized Conformal Prediction (SLCP), which uses unlabeled target data and labeled source data to stabilize localization and the threshold while maintaining marginal coverage, and shows improved stability empirically under limited calibration.

**Compliance With Llm Reviewing Policy:**

Affirmed.

**Final Justification:**

The rebuttal addresses my concerns, so I raise my score. However, the submission still requires significant revisions.

**Key Questions For Authors:**

1. The authors should better justify the key assumptions used in the analysis. In particular, it would be helpful to clarify what the constant $C$ represents in Theorem 4.2 and to discuss how the marginal coverage bound depends on the main parameters.
2. If I understand correctly, Theorem 4.5 indicates that the proposed approach improves set stability primarily in regimes where labeled source data are abundant but target calibration labels are scarce. This regime is not sufficiently motivated in the current presentation, and the practical applicability of the method may be limited without a clearer discussion of when such data availability is realistic.
3. The experimental section should include stronger baselines and broader coverage of regimes. For example, it should include additional localized conformal methods and standard baselines such as conformalized quantile regression. It would also be useful to evaluate the case where the target calibration size $n$ is large, potentially comparable to $m$, to test whether the method still helps or whether performance degrades when abundant target labels are available.

**Limitations:**

The limitation section is missing.

**Strengths And Weaknesses:**

Strengths:
1. The paper tackles an important and relatively underexplored issue in the conformal prediction literature, so the contribution is clear.
2. The theoretical guarantees are solid.

Weaknesses:
1. The methodology assumes access to labeled source data and unlabeled target data to improve set stability, but the paper does not sufficiently motivate when this setting is realistic. It would help to include a concrete motivating example early in the introduction, for instance drawn from the experimental setups and main takeaways in Section 5.2.
2. The set of experimental baselines is limited.

---

> ### Author Rebuttal · Authors · 2026-03-30
>
> Thank you for your comments. We greatly appreciate your recognition of the importance of our problem and the strength of our theoretical analysis.
> - **Weakness 1 & Question 2:** Thanks for this helpful suggestion. We agree that a concrete motivating example is helpful, and we will **add the following motivating example in the introduction in the revised version:**
>   - Consider tissue classification in medical risk stratification [5]: most patients are low-risk and yield abundant, well-labeled data, while high-risk cases are rare and often lack reliable labels due to limited follow-up or diagnostic uncertainty, resulting in scarce labeled target data but relatively abundant unlabeled target data.
>   - Our setting directly reflects this real-world scenario, **as demonstrated in our additional TissueMNIST (**TISSUE**) experiments below.**
> - **Weakness 2 & Question 3:** Thanks for this valuable feedback. We agree and will significantly expand our experimental evaluation in the revision.
>   - Specifically, we consider two score functions of local conformal methods: (1) GLCP, and (2) Conformized Quantile Regression (CQR[1]). Our method applies directly to any such score (see response to Reviewer 44Zg, Weakness 1).
>   - We further consider two semi-supervised baselines: SS[2] incorporates reconstruction loss from unlabeled data as an auxiliary feature, and SD[3] estimates conditional distributions to debias marginal calibration. Both SS and SD are applied to GLCP and CQR. Additionally, "OR" denotes the oracle result obtained with infinite calibration data.
>   - As shown in the results table, SD frequently violates marginal validity, while SS maintains validity but often increases variance. In contrast, our method consistently improves stability and reduces set size across datasets while preserving marginal validity and achieving competitive conditional coverage. **Additional marginal coverage, set size and miscoverage results are provided in our response to Reviewer nKWj.**
>
> ||Dataset|$n/m$|Raw 1|SS 1|SD 1|SL 1|OR 1||Raw 2|SS 2|SD 2|SL 2|OR 2|
> |:-:|:-:|:-:|:-:|:-:|:-:|:-:|:-:|:-:|:-:|:-:|:-:|:-:|:-:|
> |Std|CRIME|30/500|3.42|3.35|3.95|**2.92**(84.3%)|2.84||1.06|1.07|1.05|**0.89**(35.6%)|0.60|
> ||BIO|30/1000|0.36|0.52|0.22|**0.25**(52.4%)|0.15||0.30|0.34|0.27|**0.15**(85.0%)|0.12|
> ||STAR|30/500|10.02|9.11|11.72|**7.97**(29.3%)|5.20||4.45|4.89|4.22|**4.00**(14.3%)|1.30|
> ||DERMA|30/1000|1.38|1.21|1.76|**0.99**(32.0%)|0.55||0.62|0.64|0.53|**0.51**(20.9%)|0.11|
> ||TISSUE|30/1000|1.20|1.14|1.19|**0.91**(25.7%)|0.25||1.20|1.34|1.39|**0.83**(37.9%)|0.22|
>
> - **Question 1:** Thanks for this important question. We will provide following justification for these assumptions in the revised version.
>   - Assumption 4.1 holds when the model parameters are bounded and the data lie in a compact support, continuous differentiability and strict positivity of the conditional density $f(v|x;\theta)$ in both $v,\theta$. Compactness ensure the density is uniformly bounded away from zero and infinity. This implies that the conditional distribution function is Lipschitz continuous. Such assumptions are common in related work; for example, they align with Assumption 1 in [4].
>   - In Theorem 4.2, the constant $C$ is proportional to: (1) the norm bound of $\Theta$, (2) the reciprocal of the lower bound of the conditional density $f(v|x;\theta)$, and (3) the Lipschitz constant of $F(\cdot|x;\theta)$. A large parameter norm or large Lipschitz constant can amplify small perturbations in the score function, while a very small density lower bound makes the quantile estimation ill-conditioned, both leading to larger coverage deviations.
> - **Question 3:** We conducted an additional experiment on the synthetic data from Section 5.1 with $\gamma=0.9$ and $m=500$, varying $n$ to 100, 500 (comparable to $m$), while keeping other settings identical. Results show that SLCP consistently improves stability compared to other method while maintaining marginal validity, though the magnitude of improvement diminishes as the target calibration size increases. In the revised version, we will include these experiments.
>
> ||$n/m$|Raw 1|SS 1|SD 1|SL 1|OR 1||Raw 2|SS 2|SD 2|SL 2|OR 2|
> |:-:|:-:|:-:|:-:|:-:|:-:|:-:|:-:|:-:|:-:|:-:|:-:|:-:|
> |Std|100/500|1.24|1.29|1.38|**0.98**(63.9%)|0.84||0.20|0.21|0.22|**0.18**(16.2%)|0.09|
> ||500/500|0.70|0.69|0.58|**0.64**(32.4%)|0.515||0.12|0.13|0.12|**0.11**(15.0%)|0.04|
> |Marginal|100/500|0.900|0.900|0.863-|0.896|0.902||0.911|0.911|0.882|0.907|0.901|
> ||500/500|0.902|0.898|0.872-|0.896|0.899||0.897|0.899|0.882-|0.897|0.902|
> |Size|100/500|3.80|**3.63**|3.19|3.66|3.72||2.15|2.14|1.95|**2.11**|2.05|
> ||500/500|3.93|**3.60**|3.19|3.74|3.87||**1.92**|**1.92**|1.82|**1.92**|1.95|
> |Miscoverage|100/500|0.025|0.025|0.057|0.018|0.008||0.018|0.021|0.029|0.017|0.005|
> ||500/500|0.015|0.015|0.299|0.010|0.006||0.011|0.015|0.020|0.010|0.004|
>
> Reference [1]-[4] see response to Reviewer bcPa.
>
> [5] Esteva A, et al. Deep learning-enabled medical computer vision.

---

> > ### Author Rebuttal · Reviewer_Pd6D · 2026-04-02
> >
> > Thank the authors for their substantial efforts during the rebuttal and for thoughtfully addressing the concerns. A more detailed discussion of the similarities to and differences from prediction-powered inference [1], which also leverages unlabeled data to achieve a form of variance reduction, would further strengthen the paper. If feasible, including it as a baseline would make the evaluation more solid.
> >
> > [1] Angelopoulos, Anastasios N., et al. "Prediction-powered inference." Science 382.6671 (2023): 669-674.
> >
> > Thanks the authors for the detailed response about the discussions and additional experiments with PPI. I'll raise my score.

---

> > > ### Author Response · Authors · 2026-04-02
> > >
> > > Thank you for the insightful suggestion to clarify the relationship between our method and Prediction-Powered Inference (PPI). Both approaches leverage unlabeled data to improve inference through a two-stage pseudo-labeling+recalibration pipeline. However, they differ significantly in both objective and implementation.
> > >
> > > **Similarities**:
> > > - Both aim to reduce variance/uncertainty by incorporating unlabeled data.
> > > - Both methods first generate pseudo-labels for unlabeled data: PPI uses any predictor, while our approach samples from an estimated conditional distribution.
> > >
> > > **Key Differences**:
> > > - Different goal: PPI constructs more stable prediction intervals for parameters like quantile, whereas ours targets marginally valid prediction sets for individual outcome with more stable performance.
> > > - PPI reduces variance by debiasing a loss-based parameter estimator, and ours by optimizing a penalized distributional $R$-distance.
> > >
> > > **Implementation and Experiments**
> > > - **Adapt PPI to our setting**: one would replace the quantile estimator $\hat{q}\_{SL}$ with a quantile estimate $\hat{q}\_{PPI}$ obtained from PPI procedure, with predictor trained on the source labeled data.
> > > - **PPI-based method has no marginal guarantee**: the PPI-based methods construct conformal sets using quantile estimates which relies heavily on the accuracy of the underlying predictor. Thus PPI-based methods has no marginal guarantee. In contrast, our method explicitly enforces marginal validity by directly aligning the marginal distribution $\hat{F}_S^0$.
> > > - **Empirical performance**:
> > >   - We train the predictor via engression, comparable to other baselines. Below we have updated our real-data experiments to include PPI as a baseline. We will add a detailed discussion of these points in the revised manuscript.
> > >   - the empirical performance of PPI exhibits instability similar to that of SD: in certain cases (e.g., PPI-CQR on CRIME and DERMA, and PPI-GLCP on DERMA), its marginal coverage deviates from the nominal level $1-\alpha$; yet, in many settings, it achieves variance reduction compared to the base method. Nevertheless, it consistently underperforms our approach.
> > >   - More critically, although the marginal coverage of PPI-based methods remains within 2% deviation range in some cases, in many settings it exhibits a greater degree of under-coverage compared to the base method, SS, or our approaches, highlighting its lack of robustness in finite samples. This implies that directly applying PPI to construct conformal prediction sets is less stable than the SL method.
> > >
> > > ||Dataset|$n/m$|GLCP|SS-GLCP|SD-GLCP|PPI-GLCP|ours-GLCP|OR-GLCP||CQR|SS-CQR|SD-CQR|PPI-CQR|ours-CQR|OR-CQR|
> > > |:-:|:-:|:-:|:-:|:-:|:-:|:-:|:-:|:-:|:-:|:-:|:-:|:-:|:-:|:-:|:-:|
> > > |Std|CRIME|30/500|3.42|3.35|3.95|3.25|**2.92**(81.4%)|2.84||1.06|1.07|1.05|1.03|**0.89**(35.3%)|0.60|
> > > ||BIO|30/1000|0.36|0.52|0.22|0.31|**0.25**(37.2%)|0.15||0.30|0.34|0.27|0.28|**0.15**(82.6%)|0.12|
> > > ||STAR|30/500|10.02|9.11|11.72|8.73|**7.97**(21.5%)|5.20||4.45|4.89|4.22|4.40|**4.00**(13.0%)|1.30|
> > > ||DERMA|30/1000|1.38|1.21|1.76|1.10|**0.99**(32.0%)|0.55||0.62|0.64|0.53|0.53|**0.51**(20.3%)|0.11|
> > > ||TISSUE|30/1000|1.20|1.14|1.19|1.12|**0.91**(24.7%)|0.25||1.20|1.34|1.39|1.11|**0.83**(31.1%)|0.22|
> > > |Marginal|CRIME|30/500|0.895|0.904|0.866-|0.886|0.908|0.896||0.885|0.882|0.870-|0.878-|0.888|0.892|
> > > ||BIO|30/1000|0.902|0.897|0.860-|0.895|0.902|0.903||0.893|0.897|0.889|0.893|0.898|0.899|
> > > ||STAR|30/500|0.913|0.909|0.871-|0.908|0.896|0.908||0.908|0.910|0.880-|0.904|0.900|0.907|
> > > ||DERMA|30/1000|0.923|0.925|0.894|0.880-|0.908|0.899||0.916|0.914|0.906|0.879-|0.903|0.897|
> > > ||TISSUE|30/1000|0.897|0.899|0.888|0.888|0.906|0.903||0.897|0.901|0.884|0.893|0.900|0.902|
> > > |Size|CRIME|30/500|6.41|6.45|6.51|**6.20**|6.41|6.37||4.27|4.28|4.08|4.09|**4.27**|4.30|
> > > ||BIO|30/1000|1.50|1.49|1.27|**1.42**|1.46|1.42||1.55|1.59|1.52|**1.52**|**1.52**|1.52|
> > > ||STAR|30/500|56.6|54.7|48.2|53.8|**52.5**|53.7||43.2|42.8|43.1|**41.9**|42.0|41.3|
> > > ||DERMA|30/1000|2.99|2.83|2.92|2.27|**2.52**|2.07||2.00|2.09|1.87|1.85|**1.84**|1.61|
> > > ||TISSUE|30/1000|4.44|4.27|4.29|**4.26**|4.37|4.14||4.54|4.57|4.46|**4.44**|**4.44**|4.29|
> > > |Miscoverage|CRIME|30/500|0.030|0.035|0.034|0.037|0.035|0.035||0.044|0.046|0.040|0.043|0.041|0.040|
> > > ||BIO|30/1000|0.032|0.027|0.048|0.032|0.032|0.033||0.037|0.036|0.036|0.036|0.038|0.038|
> > > ||STAR|30/500|0.033|0.029|0.036|0.028|0.039|0.024||0.042|0.039|0.044|0.040|0.041|0.043|
> > > ||DERMA|30/1000|0.040|0.045|0.035|0.039|0.035|0.043||0.069|0.058|0.072|0.074|0.071|0.090|
> > > ||TISSUE|30/1000|0.053|0.054|0.052|0.053|0.057|0.060||0.054|0.057|0.048|0.052|0.057|0.060|
> > >
> > > *Thank you for your kind response and for raising your score and we're glad our revisions addressed your concerns! We’d be grateful for any additional suggestions on how we could further strengthen the paper. Thanks again for your feedback!*

---

### Official Review · Reviewer_bcPa · 2026-02-27

**Soundness:** 3
**Presentation:** 2
**Significance:** 3
**Originality:** 3
**Overall Recommendation:** 3
**Confidence:** 3

**Summary:**

The authors address the concept of prediction-set instability in conformal prediction when calibration data are limited. I think this is a meaningful job. My main concerns are as follows.

**Compliance With Llm Reviewing Policy:**

Affirmed.

**Final Justification:**

The authors' rebuttal did not fully address my concerns and exacerbated the following concerns:

1. The provided anonymous repository does not include a README or similar documentation, so I cannot assess or verify the reproducibility of the method.

2. The unavailability of the proposed method on large-scale image data exacerbates my concerns about the scope of applicability of the method.

**Key Questions For Authors:**

Could you test it on a popular large-scale classification dataset, such as ImageNet?

**Limitations:**

The author did not discuss the limitations of the proposed method.

**Strengths And Weaknesses:**

Strength:

- The paper clearly identifies that existing criteria (prediction efficiency, test-conditional coverage) are defined in expectation over calibration data and do not capture variability of set size for a single calibration sample.

Weakness:

- The experimental comparisons do not appear to have included comparisons with other semi-supervised conformal prediction methods. Such comparisons are necessary, as they introduce unsupervised information about the target domain.
- The validity of Assumption 4.1 in practical applications has not been discussed.
- The author did not provide reproducible code, making verification of the results difficult.
- “We assume transfer learning is feasible: while the covariate distributions $P_X$ and $P_{X}^{'}$ may differ substantially (potential covariate shift), the conditional distributions $P_{Y|X}$ and $P_{Y |X}^{'}$ are similar (moderate label shift).” This statement is incorrect and does not conform to the definition of label shift.

---

> ### Author Rebuttal · Authors · 2026-03-30
>
> Thank you for highlighting this key point. We appreciate your clear recognition of the paper's core motivation.
>
> - **Weakness 1:** Thank you for raising this important point.
>     -   We agree and will **expand our experimental comparison in the revised version to include semi-supervised conformal baselines that leverage unlabeled target data.**
>     -   Specifically, we consider two score functions corresponding to two representative classes of local conformal methods: (1) GLCP (representing LCP, DCP, etc.), and (2) Conformized Quantile Regression (CQR[1]), representing methods using quantile regression.
>     -   We consider two semi-supervised baselines: SS[2] (using reconstruction loss from unlabeled data as an auxiliary feature) and SD[3] (estimating conditional distributions to debias marginal calibration), applied to both GLCP and CQR. Also, "OR" represents the oracle result using infinite calibration data.
>     -   Our method naturally extends to any score function (see response to Reviewer 44Zg, Weakness 1).
>     -   The results in the following table show that SD often violates marginal validity, while SS preserves validity but typically increases variance. In contrast, our method consistently reduces variance and improves set size across datasets while maintaining both marginal validity and competitive conditional coverage; for the remaining set size and miscoverage results, **please see the tables in our response to Reviewer nKWj.**
>     - Due to time constraints, we only report new real-data results here; **synthetic results will be updated in the revised version.**
>
> ||Dataset|$n/m$|Raw 1|SS 1|SD 1|SL 1|OR 1||Raw 2|SS 2|SD 2|SL 2|OR 2|
> |:-:|:-:|:-:|:-:|:-:|:-:|:-:|:-:|:-:|:-:|:-:|:-:|:-:|:-:|
> |Std|CRIME|30/500|3.42|3.35|3.95|**2.92**(84.3%)|2.84||1.06|1.07|1.05|**0.89**(35.6%)|0.60|
> ||BIO|30/1000|0.36|0.52|0.22|**0.25**(52.4%)|0.15||0.30|0.34|0.27|**0.15**(85.0%)|0.12|
> ||STAR|30/500|10.02|9.11|11.72|**7.97**(29.3%)|5.20||4.45|4.89|4.22|**4.00**(14.3%)|1.30|
> ||DERMA|30/1000|1.38|1.21|1.76|**0.99**(32.0%)|0.55||0.62|0.64|0.53|**0.51**(20.9%)|0.11|
> ||TISSUE|30/1000|1.20|1.14|1.19|**0.91**(25.7%)|0.25||1.20|1.34|1.39|**0.83**(37.9%)|0.22|
> |Marginal|CRIME|30/500|0.895|0.904|0.866-|0.908|0.896||0.885|0.882|0.870-|0.888|0.892|
> ||BIO|30/1000|0.902|0.897|0.860-|0.902|0.903||0.893|0.897|0.889|0.898|0.899|
> ||STAR|30/500|0.913|0.909|0.871-|0.896|0.908||0.908|0.910|0.880-|0.900|0.907|
> ||DERMA|30/1000|0.923|0.925|0.894|0.908|0.899||0.916|0.914|0.906|0.903|0.897|
> ||TISSUE|30/1000|0.897|0.899|0.888|0.906|0.903||0.897|0.901|0.884|0.900|0.902|
>
> - **Weakness 2:** Thank you for this observation.
>   - Assumption 4.1 holds under mild regularity conditions: for a neural network with bounded parameters and data supported on a compact set, if the conditional density $f(v|x;\theta)$ is continuously differentiable and strictly positive in both $v$ and $\theta$, compactness guarantees uniform upper and lower bounds on the density, which in turn imply Lipschitz continuity of the corresponding distribution function.
>   - Hence, Assumption 4.1 is reasonable in practical settings.
>   - Similar assumptions are made in many literature, like in Assumption 1 of [4].
>   - **We will add the discussion and justification of Assumption 4.1 in the revised version.**
>
> - **Weakness 3:** Thank you for this comment. We confirm that the code will be made publicly available on GitHub upon acceptance to ensure full reproducibility.
>
> - **Weakness 4:** Thank you for catching this imprecision. This is indeed a typo in terminology. We do not assume label shift but rather that the conditional distributions $P_{Y|X}$ and $P_{Y|X}'$ are similar across domains, i.e., conditional shift is mild, while allowing arbitrary covariate shift in $P(X)$. **We will correct this wording in the revised manuscript.**
>
> - **Question 1:** Thank you for the suggestion.
>   - Our additional experiments include TissueMNIST (**TISSUE**) from the MedMNIST benchmark, the largest 2D subset with over 200K samples, which serves as a suitable large-scale classification dataset for our setting.
>   - As shown in [5], standard baselines achieve relatively low accuracy on this dataset, making it a meaningful testbed for evaluating conformal prediction stability. In each replication, we retrain the base predictor on the full training set and specifically target stable and accurate conditional coverage on minority classes. Results are reported in the table above.
>
> [1] Romano Y, et al. Conformalized Quantile Regression.
>
> [2] Seedat N, et al. Improving Adaptive Conformal Prediction Using Self-Supervised Learning.
>
> [3] Wen M, et al. Semi-supervised distribution learning.
>
> [4] Guan L. Localized conformal prediction: a generalized inference framework for conformal prediction.
>
> [5] Yang J, et al. MedMNIST v2-A large-scale lightweight benchmark for 2D and 3D biomedical image classification.

---

> > ### Author Rebuttal · Reviewer_bcPa · 2026-04-03
> >
> > Thanks for the author's reply. I still have the following unresolved questions:
> >
> > 1. The authors have not provided sufficient evidence for reproducibility.
> >
> > 2. Compared with ImageNet, TissueMNIST is considerably smaller. This therefore does not alleviate my concerns about the scope of applicability of the proposed method.

---

> > > ### Author Response · Authors · 2026-04-03
> > >
> > > Thank you for your thoughtful comment. We have prepared and shared our code at the following anonymous link: https://anonymous.4open.science/r/SLCP-98CC/.
> > >
> > > Our method is specifically designed for scenarios where labeled calibration data from the target distribution is scarce, a setting in which marginal quantile estimation of the conformity score becomes unstable. When sufficient calibration data is available, this instability vanishes, as the quantile estimate is already well-determined; hence, all our experiments focus on the small-calibration regime, which best reflects the problem we aim to address.
> > >
> > > Regarding dataset scale: in our context, the "difficulty" of stabilization is not determined by raw dataset size alone, but by how inaccurate the base predictor is, i.e., the harder the underlying learning problem, the noisier the scores, and the greater the instability. Among the MedMNIST benchmarks, TissueMNIST is both the largest and the one with the lowest baseline accuracy, making it the most challenging case for conformal calibration within this suite. We deliberately selected it to demonstrate the applicability of our method under adverse conditions. While ImageNet is indeed larger, the challenges posed under sparse calibration are largely comparable across datasets, as they stem from the limited number of labeled calibration points rather than the total size of the source or unlabeled data.
> > >
> > > We acknowledge that ImageNet experiments involve additional complexity, and we are actively working on them. We will update the results as soon as they are ready.

---

### Official Review · Reviewer_44Zg · 2026-03-12

**Soundness:** 3
**Presentation:** 2
**Significance:** 2
**Originality:** 3
**Overall Recommendation:** 4
**Confidence:** 3

**Summary:**

The paper addresses the generalized local conformal prediction which first estimates a conditional score distribution over every point, then for each evidence (points in the calibration set) it defines a linear mapping over the CDF of the score within the base estimated (conditional) distribution.  Then a CP is applied over this first-order score. The authors are arguing that this framework is sensitive to the randomness in calibration set, meaning that by redrawing the calibration set, the set size can vary a lot.

Their approach is to reduce this variance through another labeled data source (which potentially can have distribution shift), and an unlabeled set of datapoints from the same domain. The general CDF can be decomposed into a dataset sampling, and a conditional distribution over the true score. The authors replace the internal expectation with an estimator trained on data from a nearby distribution, and use the unlabeled data for the outer expectation. This works since the inner expectation now has an estimator which makes it independent of the labels.

Basically their method assumes that there is an estimator of the conditional score CDF (that in high level receives x, and s, and returns a quantile) by which GLCP is providing a guarantee. Now the authors finetune that estimator with another set of labeled data (not potentially coming from the same distribution) and then with an unlabeled data they estimate the same CDF over the new finetuned estimator.

**Compliance With Llm Reviewing Policy:**

Affirmed.

**Final Justification:**

I thank the author for responding to my questions. Still the paper can be rewritten in an easier-to-read way. The response to reviewer nkwj convinced me about the position of the paper w.r.t. other baselines.

**Key Questions For Authors:**

1. I think the authors’ method remains valid as long as they show that the Eq. 6 can be solved with a global, and not a local minima. Can the authors elaborate on the approach to solve such a problem ensuring that the result would at least find the GLCP? Is it guaranteed that the solution of GLCP is always in the function family even if the family is parametric? Also I am not sure when Assumption 4.1 is valid.

2. Basically what the authors rely on is a Lipschitz continuity of the function class which is used both in GLCP, and SLP. I am not sure in what cases this assumption remains valid. If not it could be that the conditional distributions estimators diverge significantly based on the covariate shift between the two distributions.

3. I am confused reading the result tables. So I assume if the objective is to reduce the variance in set size the authors result is always better than GLCP, and worse than SCP. I think additionally I should note that the conditional coverage is higher in SCP. Therefore the SCP should not be compared with SLCP. Is that so? If that is the case I ask the authors to change the color of SCP or make the conditional miscoverage of SCP bold/red, or in any other ways point out that the methods are not comparable.

**Limitations:**

Not sure, but from what I understood the method only works on a small collection of estimators, which means that it can not be applied on complicated models and data types.

**Strengths And Weaknesses:**

I think the problem is in general interesting and the motivation which leads to this paper is very important. However in the sections after the introduction I realized that the motivation is applied in a very specific class of algorithms.

**Weaknesses**

1. I think the message in the introduction and even in abstract is more general than the baseline framework the authors improved. Mainly the authors are improving GLCP, while by reading the introduction, I thought that the authors are trying to deal with the general issue of calibration set instability in any CP approach.

2. I think the point of writing an algorithm is to deliver a quick intuition of the overall procedure, to someone reading it before reading the paper entirely. The current algorithm 1 is not doing so. Maybe the authors need to rewrite this algorithm at least by pointing to the relevant procedure, e.g. for steps 3, and 4.

---

> ### Author Rebuttal · Authors · 2026-03-30
>
> Thanks for recognizing the importance of the problem and the motivation behind our work.
> - **Weakness 1:**
>   - While the introduction emphasizes the general issue of calibration set instability in CP, we focus on local CP methods like GLCP because they are particularly susceptible to such instability (as supported by our simulations; see also the table in response to Reviewer nKWj). Our method stabilizes the marginal distribution estimate of a given conformity score using unlabeled target data (via $\hat{F}_S^1$) while preserving marginal coverage by aligning distributions as in eq (6). **The revised introduction will clarify that GLCP-type methods are our primary focus due to their heightened sensitivity to calibration set variability, while noting that our approach extends naturally to general conformity scores.**
>   - For any CP method with $S(x, y)$, one can:
>     - Obtain $\hat{F}_S^0(s)=n^{-1}\sum\_{i=1}^n1(S(X_i,Y_i)\leq s)$ based on labeled calibration set $\mathcal{L}_n$.
>     - Estimate the conditional distribution $\hat{F}_{S|X}$ using source data $\mathcal{L}'_N$.
>     - Generate a candidate distribution $\hat{F}_S^1(s;\tilde{F})=m^{-1}\sum\_{j=1}^mE\\{1(\tilde{S}_j\leq s)\\}$ for $\tilde{S}_j\sim\tilde{F}(\cdot|\tilde{X}_j)$ via unlabeled data $\mathcal{U}_m$.
>     - Align $\hat{F}_S^1(\cdot;\tilde{F})$ with the reference $\hat{F}_S^0(\cdot)$ and penalize the distance between $\tilde{F}$ and $\hat{F}\_{S|X}$:
>       - $\tilde{F}_{S|X}={\rm argmin}~d(\hat{F}_S^0(\cdot),\hat{F}_S^1(\cdot;\tilde{F}))+\lambda R(\tilde{F},\hat{F}\_{S|X})$ for $\tilde{F}\in\mathcal{F}$ corresponds to eq (6) of the main text.
>     - The only modification required is replacing $\hat{F}_{V|X}$ with $\hat{F}\_{S|X}$ and original score with $S(x,y)$ in our algorithm.
>   - The marginal coverage result (Theorem 4.2) continues to hold after replacing the original score $V$ and the conditional distribution of $V|X$ with the new score $S$ and $S|X$. Similarly, Lemma 4.4 and Theorem 4.5 remain valid when $\hat{F}_{V|X}$ is replaced with the pre-specified score $S(x,y)$.
>   - **We will add this generalization to the appendix, and our supplementary experiment "ours-CQR" was already implemented using this generalization.**
> - **Weakness 2:** Thanks for the suggestion. We'll revise Algorithm 1 to clearly reference the key steps (e.g., marginal estimation in Step 3 and alignment in Step 4) for better intuition.
> - **Question 1:** Thank you for the question. In practice, we discretize the distribution using 50 quantiles and minimize the $R$ distance between two distributions in eq (6). Our model has 1500 tunable parameters that far exceeding $50$, so when $\lambda \to 0$, the optimization becomes an underdetermined system that can achieve zero loss. Empirically, gradient descent with $\lambda=0$ drives the loss to $0$, and the resulting quantile estimate $\hat{q}$ matches that of GLCP, thus recovering GLCP exactly in this case. To summarize, since our function class is sufficiently expressive (overparameterized), it contains the GLCP solution even in parametric settings.
> - **Question 1 (assumption validity) and 2:** The revised version will include the following justification:
>   - For a neural network with all parameters bounded and data supported on a compact set, if the conditional density \$f(v\|x;\theta) \$ is continuously differentiable and non-zero in both $v$ and $\theta$, then compactness implies uniform upper and lower bounds, which also ensure Lipschitz continuity. Thus Assumption 4.1 is valid.
>   - The Lipschitz assumption on $x$ is equivalent to the density being bounded above. As discussed, this holds in a broad range of commonly used settings, such as Assumption 1 in [1]; For assumption on $\theta$, consider neural network $f(x;\theta)$ composed of affine transformations (in $\theta$) and Lipschitz activation functions. When input $x$ and parameters $\theta$ lie in a bounded space, this composition is Lipschitz continuous on $\theta$. This property underlies the stability analysis in Section 3 of [2], which relies on the loss (and thus the model output) being Lipschitz in the parameters.
>
> - **Question 3:** Thanks for raising this concern, and sorry for the confusion.
>   - Our primary focus is on the instability of local conformal methods. SCP, while stable, achieves significantly lower conditional coverage and serves only as a non-adaptive baseline. We do not consider its stability comparable to GLCP/SLCP. To avoid misleading comparisons, **we will clarify this in the revised tables and have removed SCP from our updated real-data experiments.**
>   - The revised version will include additional experiments by incorporating more local score functions and semi-supervised conformal baselines; **see our response to Reviewer nKWj.**
>
> [1] Guan L. Localized conformal prediction: a generalized inference framework for conformal prediction.
>
> [2] Hardt M, et al. Train faster, generalize better: Stability of stochastic gradient descent.

---

> > ### Author Rebuttal · Reviewer_44Zg · 2026-04-03
> >
> > Thanks for the reply. I still strongly suggest a better flow in text, and better interpretation of results in the camera-ready. But for now I increase my score.

---

> > > ### Author Response · Authors · 2026-04-04
> > >
> > > Thank you for your response and for raising your score!  We agree with your suggestions and will improve the text flow and result interpretation in the revised version.
> > >
> > > Thanks again for your valuable feedback!

---

### Official Review · Reviewer_nKWj · 2026-03-18

**Soundness:** 3
**Presentation:** 3
**Significance:** 3
**Originality:** 2
**Overall Recommendation:** 5
**Confidence:** 4

**Summary:**

The paper addresses a common point of concern with conformal methods. To succeed in practice, we need calibration data of sufficient size, which is not always at hand. Quite often, especially in very applied settings in specific domains where data collection might be quite expensive, calibration set sizes are quite small (there is a premium on every point that you can afford), which make the prediction sets/intervals unreliable. Generally, practitioners rely on all sorts of hacks to improve reliability. This is something not usually discussed in the literature, except for a small literature in localization, because it is not considered a research problem. The idea in localization (e.g. localized conformal by Guan et al, LVD by Lin et al, which makes Guan's procedure "learnable" to mitigate neighborhood sparsity) is to include a localization in the calibration process, but neighborhood sparsity could get even more pronounced (could be mitigated somewhat by learning, but the problem remains). The paper focuses on this concern via the language of set stability. First the sensitivity of PIs/PSs to calibration data is formalized by this notion of set stability, and then a transfer learning-based approach is proposed to improve stability. Marginal coverage results and improvements in set stability are shown. Experiments are not very satisfactory in terms of design of baselines, but are sufficient to illustrate the main story of the paper.

**Compliance With Llm Reviewing Policy:**

Affirmed.

**Key Questions For Authors:**

See above.

**Limitations:**

Yes

**Strengths And Weaknesses:**

Strengths: The paper is well-written and presented. It is easy to follow the main motivation and the procedure. The theoretical results look reasonable and sound (although I have not verified the proofs fully). The approach presented is natural, and a transfer learning approach to improve stability (not specifically in the conformal context) is the de facto go to.

Weaknesses: As mentioned above. The main issue is with the experiments. The experiments are toy and don't use any baselines. These could be constructed by using various variants of local conformal methods and making the synthetic data even more adversarial for them. It would be good to see the failure cases for an ensemble of those methods illustrated. Nevertheless, the experiments are sufficient artifacts to support the main theory and narrative of the paper.

Minor: The usage of the "labeled source data" "labeled target data" in the paper is a bit confusing. Takes a bit of back and forth at times to clarify what it is intending to say. Further, it would also be good to be a bit more exhaustive in referencing the literature. The idea of set stability in the conformal context might be somewhat newer, but it is an old concept in learning theory, and also in unsupervised learning (see the monograph by von Luxburg) and semi-supervised learning. A few lines connecting to that lineage would be useful for readers.

---

> ### Author Rebuttal · Authors · 2026-03-30
>
> Thank you for your positive feedback on the clarity of our writing, the soundness of our theoretical results, and the naturalness of our transfer learning approach.
>
> -   Regarding concern about experiments, we agree and will extend our evaluation in the revised version.
>     -   **We will consider two score functions** corresponding to two representative classes of local conformal methods:
>
>         - GLCP in the origin paper, representing methods such as LCP and DCP;
>
>         - Conformized Quantile Regression (CQR[1]), representing methods that utilizes quantile regression.
>     -   Along with our proposed SLCP method, **we consider two baselines: self-supervised (SS) [2] and semi-distribution (SD) [3], applied to both GLCP and CQR scores, with “OR” denoting the oracle benchmark based on infinite calibration data.** SS leverages unlabeled data by incorporating reconstruction loss as an auxiliary feature, while SD estimates the conditional distribution from labeled samples to produce a debiased marginal estimate.
>     -   It is worth noting that our method generalizes directly to arbitrary conformity scores (see response to Reviewer 44Zg, Weakness 1).
>
>     - Results show that SD fails to guarantee marginal validity in many cases. While SS ensures marginal validity, it often increases variance, though it improves conditional coverage in some cases. In contrast, our method significantly improves stability and size across almost all datasets while maintaining marginal validity and conditional coverage.
>
>     - Since different score constructions lead to varying conditional performance, we only compare base method (use only target labeled data), SS, SD, and our approach within groups sharing the same score function.
>
>     - Since SD sometimes fails to guarantee marginal validity, we bold the best results among base method, SS, and ours in the Std and Size columns. Additionally, in the Std results, we report the percentage improvement of ours over base method and SS relative to the oracle (OR) to quantify how much closer our method brings performance to the oracle. Entries deviating from the nominal coverage by more than 2% are marked with a -.
>
>     - Due to time constraints, we only report new real-data results here; **synthetic results will be updated in the revised version.**
>
> ||Dataset|$n/m$|GLCP|SS-GLCP|SD-GLCP|ours-GLCP|OR-GLCP||CQR|SS-CQR|SD-CQR|ours-CQR|OR-CQR|
> |:-:|:-:|:-:|:-:|:-:|:-:|:-:|:-:|:-:|:-:|:-:|:-:|:-:|:-:|
> |Std|CRIME|30/500|3.42|3.35|3.95|**2.92**(84.3%)|2.84||1.06|1.07|1.05|**0.89**(35.6%)|0.60|
> ||BIO|30/1000|0.36|0.52|0.22|**0.25**(52.4%)|0.15||0.30|0.34|0.27|**0.15**(85.0%)|0.12|
> ||STAR|30/500|10.02|9.11|11.72|**7.97**(29.3%)|5.20||4.45|4.89|4.22|**4.00**(14.3%)|1.30|
> ||DERMA|30/1000|1.38|1.21|1.76|**0.99**(32.0%)|0.55||0.62|0.64|0.53|**0.51**(20.9%)|0.11|
> ||TISSUE|30/1000|1.20|1.14|1.19|**0.91**(25.7%)|0.25||1.20|1.34|1.39|**0.83**(37.9%)|0.22|
> |Marginal|CRIME|30/500|0.895|0.904|0.866-|0.908|0.896||0.885|0.882|0.870-|0.888|0.892|
> ||BIO|30/1000|0.902|0.897|0.860-|0.902|0.903||0.893|0.897|0.889|0.898|0.899|
> ||STAR|30/500|0.913|0.909|0.871-|0.896|0.908||0.908|0.910|0.880-|0.900|0.907|
> ||DERMA|30/1000|0.923|0.925|0.894|0.908|0.899||0.916|0.914|0.906|0.903|0.897|
> ||TISSUE|30/1000|0.897|0.899|0.888|0.906|0.903||0.897|0.901|0.884|0.900|0.902|
> |Size|CRIME|30/500|**6.41**|6.45|6.51|**6.41**|6.37||4.27|4.28|4.08|**4.27**|4.30|
> ||BIO|30/1000|1.50|1.49|1.27|**1.46**|1.42||1.55|1.59|**1.52**|**1.52**|1.52|
> ||STAR|30/500|56.6|54.7|48.2|**52.5**|53.7||43.2|42.8|39.5|**42.0**|41.3|
> ||DERMA|30/1000|2.99|2.83|2.92|**2.52**|2.07||2.00|2.09|1.87|**1.84**|1.61|
> ||TISSUE|30/1000|4.44|**4.27**|4.29|4.37|4.14||4.54|4.57|4.46|**4.44**|4.29|
> |Miscoverage|CRIME|30/500|0.030|0.035|0.034|0.035|0.035||0.044|0.046|0.040|0.041|0.040|
> ||BIO|30/1000|0.032|0.027|0.048|0.032|0.033||0.037|0.036|0.036|0.038|0.038|
> ||STAR|30/500|0.033|0.029|0.036|0.039|0.024||0.042|0.039|0.044|0.041|0.043|
> ||DERMA|30/1000|0.040|0.045|0.035|0.035|0.043||0.069|0.058|0.072|0.071|0.090|
> ||TISSUE|30/1000|0.053|0.054|0.052|0.057|0.060||0.054|0.057|0.048|0.057|0.060|
>
> -  **The usage of the "labeled source data" abd "labeled target data":** We apologize for the confusion. To clarify, labeled source data refers to $\mathcal{L}_N^\prime$ and labeled target data refers to $\mathcal{L}_n$. We will make these definitions explicit in the introduction.
>
> - **Set stability in learning theory:** Thank you for the suggestion. We agree and will add a discussion connecting our concept of set stability to the classical stability analysis in learning theory, specifically referencing the training conditional nature of stability measures as seen in clustering (e.g., von Luxburg's monograph).
>
> [1] Romano Y, et al. Conformalized Quantile Regression.
>
> [2] Seedat N, et al. Improving Adaptive Conformal Prediction Using Self-Supervised Learning.
>
> [3] Wen M, et al. Semi-supervised distribution learning.

---

> > ### Author Rebuttal · Reviewer_nKWj · 2026-04-04
> >
> > Mostly resolves most of my questions. I will keep my (already positive) score.

---

> > > ### Author Response · Authors · 2026-04-05
> > >
> > > We sincerely thank you for the thoughtful feedback and for acknowledging that our responses have largely addressed the concerns. We greatly appreciate your supportive stance and positive evaluation of our work.

---

### Decision · Program_Chairs · 2026-04-30

**Decision:**

Accept (regular)

**Comment:**

This paper is on the topic of conformal prediction for uncertainty quantification and addresses a critical issue: the size of calibration data needed for the methods to succeed. Localization during conformal calibration considers this issue but neighborhood sparsity can impact its performance.  This paper tries to address this concern using the concept of set stability:  sensitivity of uncertainty sets to calibration data is formalized through set stability and a transfer learning based approach is developed to improve stability. Experimental evaluation (somewhat limited) show marginal coverage results and improvements in set stability.

All reviewers' appreciated the overall contribution and its importance, but also raised concerns about insufficient experimental evaluation (some more strongly than others). Authors' were quite responsive during the rebuttal phase and answered most questions. One reviewer wants to see experiments on large-scale image data.

Having gone through the paper myself, I agree with reviewr nKWj that experimental results are sufficient to validate the main claims of the paper.

Given the importance of the studied problem (less studied) and the paper's theoretical treatment, I support accepting the paper (can increase awareness in the CP community and result in follow up work among both researchers' and practitioners'). I strongly encourage the authors' to incorporate all the review comments / discussion to improve the final paper (especially make it easy to read, positioning of the paper w.r.t prior work, and improving experimental evaluation as much as possible) to make the paper useful to the CP community.